# Temporally restricted activation of IFNβ signaling underlies response to immune checkpoint therapy in mice

Rachael M. Zemek [1,2,3,8], Wee Loong Chin[3,4,5,8], Vanessa S. Fear [1,2,3], Ben Wylie [3], Thomas H. Casey[1,2], Cath Forbes[1,2,3], Caitlin M. Tilsed[1,2], Louis Boon[6], Belinda B. Guo [7], Anthony Bosco[3], Alistair R. R. Forrest[7], Michael J. Millward[4,5], Anna K. Nowak [1,4,5], Richard A. Lake [1,2], Timo Lassmann [3,9] ✉ & W. Joost Lesterhuis [1,2,3,9] ✉

The biological determinants of the response to immune checkpoint blockade (ICB) in cancer remain incompletely understood. Little is known about dynamic biological events that underpin therapeutic efficacy due to the inability to frequently sample tumours in patients. Here, we map the transcriptional profiles of 144 responding and non-responding tumours within two mouse models at four time points during ICB. We find that responding tumours display on/fast-off kinetics of type-I-interferon (IFN) signaling. Phenocopying of this kinetics using time-dependent sequential dosing of recombinant IFNs and neutralizing antibodies markedly improves ICB efficacy, but only when IFNβ is targeted, not IFNα. We identify Ly6C[+]/CD11b[+] inflammatory monocytes as the primary source of IFNβ and find that active type-I-IFN signaling in tumour-infiltrating inflammatory monocytes is associated with T cell expansion in patients treated with ICB. Together, our results suggest that on/fast-off modulation of IFNβ signaling is critical to the therapeutic response to ICB, which can be exploited to drive clinical outcomes towards response.

The response to immune checkpoint blockade (ICB) in cancer is highly variable, with a majority of patients experiencing disease progression. Although the targets of ICB antibodies are known, the downstream therapeutic effector mechanisms are incompletely understood[1,2]. While specific aspects of the pre-treatment tumour microenvironment, such as PD-L1 expression, immune cell infiltration[3] or tumour mutation burden[4] have been shown to correlate with response[5], none of these biomarkers are sufficiently robust to guide clinical decisions regarding treatment across cancers[6] nor has their characterization yet

resulted in approved treatments that improve the efficacy of ICB[7]. As a consequence, the development of novel combination therapies to improve outcomes is mainly empiric[8].

When perturbing complex systems, some important effects may only become apparent over time. For example, in the course of an immune response, some inflammatory mediators need to be switched off, not only to resolve inflammation but also to mediate a transition from innate to adaptive immunity[9]. Similar time-dependent mechanisms may underpin an effective anti-tumour immune response[6].

[1]National Centre for Asbestos Related Diseases, Nedlands, WA 6009, Australia. [2]School of Biomedical Sciences, University of Western Australia, Crawley, WA 6009, Australia. [3]Telethon Kids Institute, University of Western Australia, Nedlands, WA 6009, Australia. [4]Department of Medical Oncology, Sir Charles Gairdner Hospital, Nedlands, WA 6009, Australia. [5]School of Medicine, University of Western Australia, Crawley, WA 6009, Australia. [6]Polpharma Biologics, Yalelaan 46, Alexander Numan Building, 3584 CM Utrecht, The Netherlands. [7]Harry Perkins Institute of Medical Research, QEII Medical Centre and Centre for Medical Research, The University of Western Australia, Nedlands, Perth, WA 6009, Australia. [8]These authors contributed equally: Rachael M. Zemek, Wee Loong Chin.[9]These authors jointly supervised this work: Timo Lassmann, W. Joost Lesterhuis. ✉e-mail: timo.lassmann@telethonkids.org.au; willem.lesterhuis@uwa.edu.au

However, it is not possible to identify these dynamic events in human studies due to the difficulty to sample the same tumour at multiple time points during treatment, usually limiting the number of samples to two; one pre-treatment and one on-treatment sample several weeks after start of ICB[10,11]. In addition, it is exceedingly difficult to identify discrete biological differences between responding and non-responding patient tumours due to inter-individual variation in germline and cancer genetics, tumour microenvironment composition and environmental influences[12–15].

For these reasons, we optimized murine models with bilateral tumours, derived from syngeneic cancer cell lines[7]. In these models, the response to ICB with antibodies against CTLA4 and PD-L1 is symmetric, leading to either a bilateral response or a failure to respond in both tumours[7,16,17]. This means we can remove one tumour for molecular analysis while tracking the therapeutic fate of the other. Because both responders and non-responders are exposed to identical treatments, this enables the characterization of the time-dependent response to ICB in entire tumours in a highly homogenous background. These bilateral models have been used previously to identify transcriptomic ICB response signatures in the early tumour microenvironment, which was extensively validated in separate patient cohorts of bladder cancer and melanoma patients treated with ICB[7,17]. In addition, our results using these models have been independently validated by other research groups in preclinical models and patient cohorts with various tumour types, underscoring their translational relevance[18–21].

Here, we aimed to discover time-critical events in the tumour microenvironment that underlie ICB efficacy. We characterise dynamic changes in gene regulatory networks associated with response to ICB and use that information to rationally develop new schedule-dependent combination therapies.

## Results

### On/fast-off dynamics in IFN signalling are associated with response to ICB

To map the dynamic processes underlying the response to ICB, we utilized our bilateral tumour model to remove responsive and non-responsive tumours at 1 h prior and at 2-, 4- and 6-days following administration of anti-CTLA4/anti-PD-L1 therapy (Fig. 1a, b and Supplementary Fig. 1) and examined the transcriptomes of these tumours using RNA-sequencing. To avoid bias towards one tumour type, we utilized two different tumour models, AB1 mesothelioma and Renca renal cell carcinoma, and explored dynamic trends that were consistently differentially regulated between responders and non-responders in both models.

We first assessed whether there were differences in cellular composition between responders and non-responders at each time point using CIBERSORTx analysis (Fig. 1c and Supplementary Fig. 1)[22]. Renca and AB1 tumours showed a large difference in cellular makeup, yet they displayed a similar therapeutic response to ICB, underscoring that cellular composition alone is not a robust predictor. Furthermore, although some differences between responders and non-responders were observed, none of these differences were consistent between the

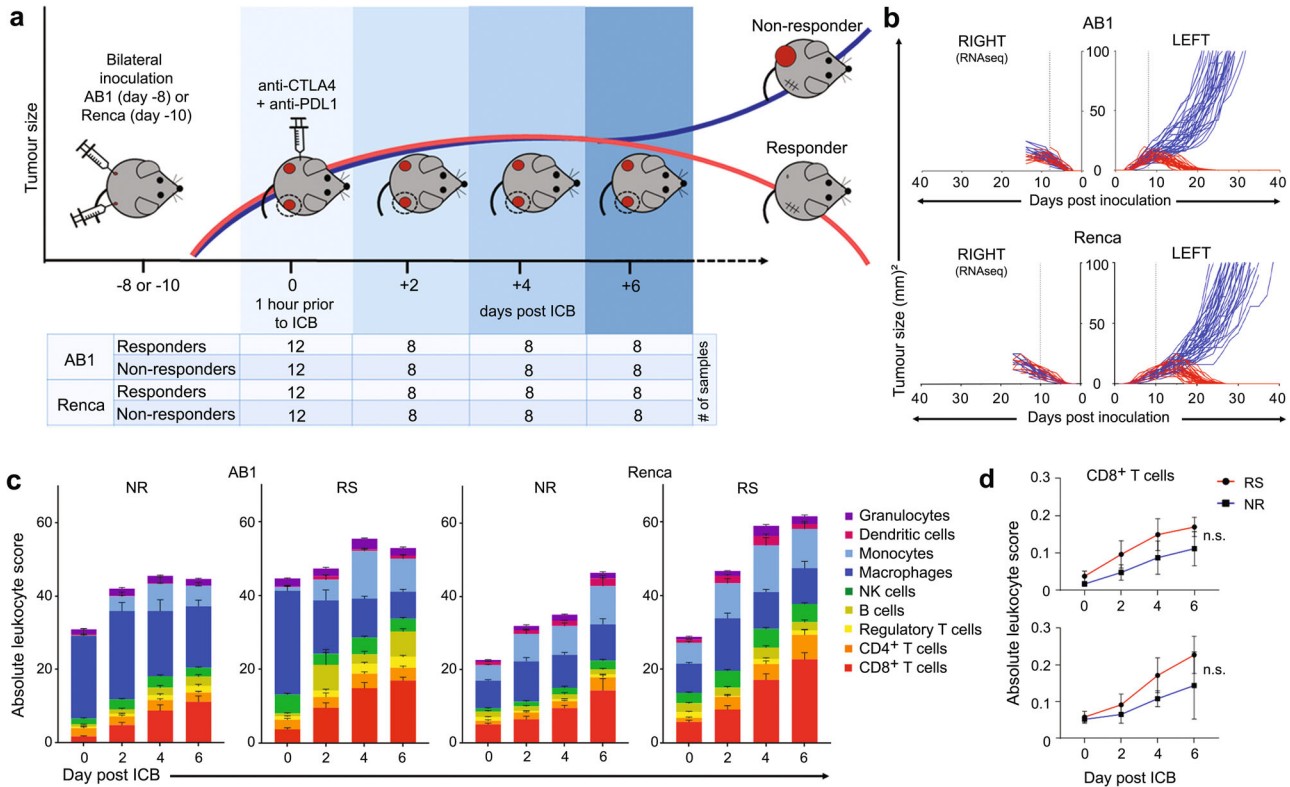

**Fig. 1 | Dual tumour models allow time-course analyses of tumours from responders and non-responders to ICB, demonstrating increased lymphoid cellular infiltration following treatment, irrespective of response. a** One tumour from mice with bilateral AB1 or Renca tumours was harvested for RNAseq either 1 h prior to, or 2, 4 or 6 days after ICB, whilst the remaining tumour was monitored for response (n = 144 biologically independent samples from 19 independent experiments). **b** Growth curves of the removed right and remaining left tumour, allowing classification of the responder (red) or non-responder (blue) to ICB. **c** CIBERSORTx cell deconvolution analysis of responder (RS) and non-responder (NR) tumours over time. Error bars represent standard deviation. **d** CD8+ T cell score in responders (RS) and non-responders (NR) over time in the AB1 (top) and Renca (bottom) models. Statistical analysis by one-way ANOVA with Benjamini-hochberg correction for multiple comparisons. Data presented as mean ± standard deviation. The number of samples per model/timepoint/group is depicted in (**a**). Source data are provided in the Source Data file.

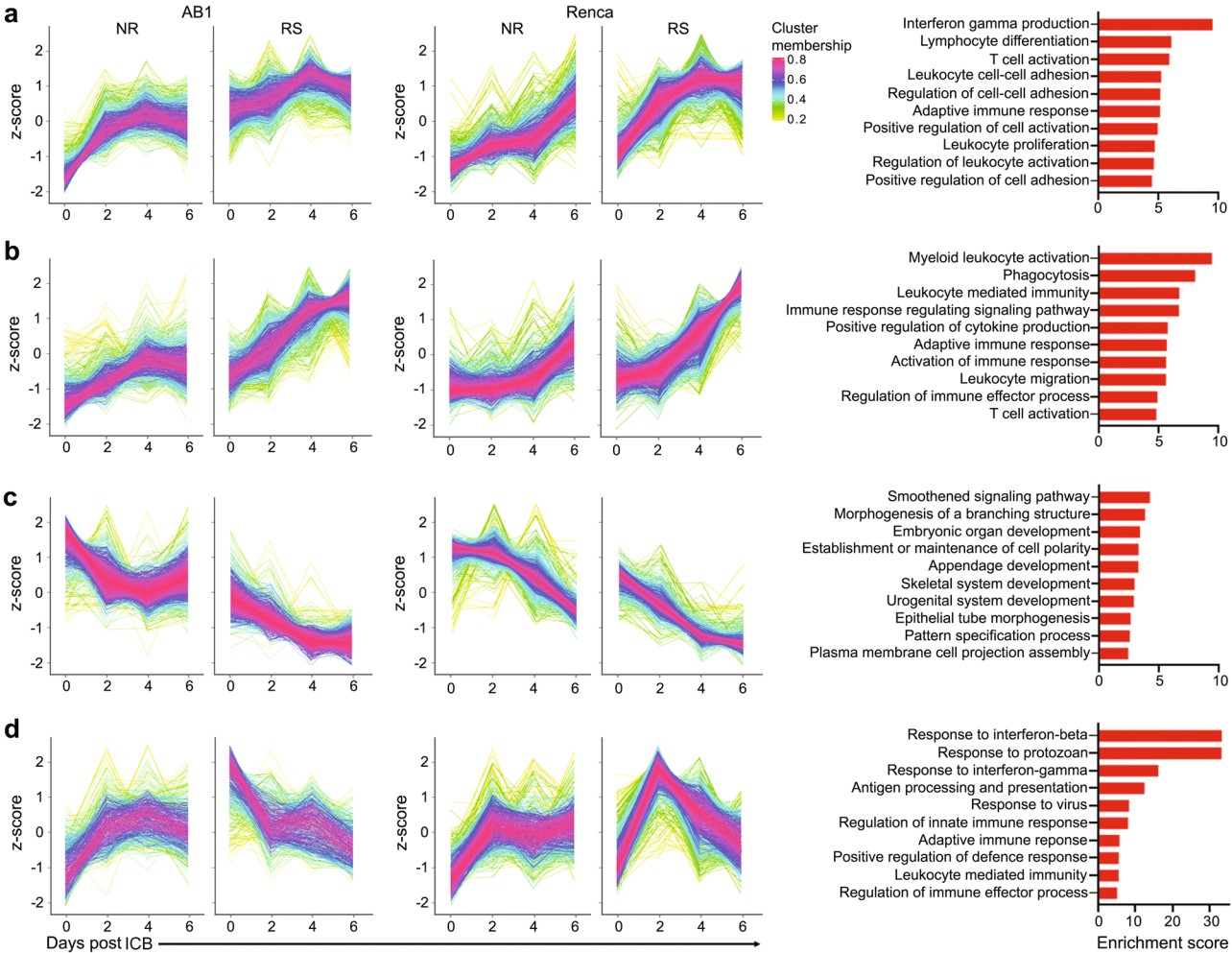

**Fig. 2 | Responders and non-responders have similar reactions to ICB, but responders shut off genes related to IFNβ signalling.** TCseq analysis was used to cluster genes with similar expression over time, identifying clusters shared by both AB1 and Renca. Gene expression over time and pathway analysis of overlapping genes between AB1 and Renca was performed for cluster 1 (**a**), cluster 2 (**b**), cluster 3 (**c**) and cluster 4 (**d**). Gene lists are provided in Supplementary Data File 3. Source data are provided in the Source Data file.

two models, except for a significantly higher proportion of NK cells in responders prior to treatment, which has been reported previously[17]. We observed an increase in CD8+ T cells after ICB, consistent with CD8+ T cell gene signatures as predictors of response in patient samples reported in the literature[23–25]. Although CD8+ T cell infiltration was more prominent in responders (Fig. 1d), it was not significantly different between responders and non-responders at any time point. As the phenotype of the cells could be a factor, particularly the expression of other checkpoints, we analysed the bulk RNAseq data to test whether responders had lower expression of checkpoints compared to non-responders. We found that checkpoint markers *Tim3, Lag3, Ox40, Icos, Ctla4, Pd1, Pdl1 and Pdl2* gene expression was higher in responders and increased after ICB, suggesting that T cell activation state rather than their numbers per se were more predictive of a response (Supplementary Fig. 2).

To understand gene expression kinetics during treatment with ICB, we clustered genes based on their dynamic expression using TCSeq[26] and analysed the resulting clusters for enriched biological pathways. We discovered four clusters that showed consistent time-dependent behaviour between the two models. Clusters 1 and 2 contained genes associated with activation of myeloid cells and T cells, including IFNγ production (Fig. 2a, b). The expression of these genes gradually increased over time, and the trend was the same in both responders and non-responders (Supplementary Fig. 3), albeit to a

greater magnitude in responders, which was in agreement with the CIBERSORT results (Fig. 1d). Genes associated with cancer cell signalling (cluster 3) decreased in expression over time, but again in both responders and non-responders (Fig. 2c and Supplementary Fig. 3). In contrast, cluster 4, demonstrated a kinetic profile that was strikingly different between responders and non-responders (Fig. 2d and Supplementary Fig. 3). Cluster 4 contained genes associated with IFN signalling, which showed a gradual increase in expression over time in non-responders, while in responders it was initially highly expressed, followed by a rapid decrease in both AB1 and Renca (Fig. 2d).

To understand the transcriptional regulation of these genes, we constructed gene regulatory networks in our ICB responders and non-responders, using the GENIE3 algorithm (Supplementary Fig. 4a)[27]. For each gene, the algorithm calculates an importance score reflecting the inferred effect of the gene on all other genes. A high importance score denotes a strong effect of a gene (putative regulator) on the dynamics of expression of a downstream gene (target) in the network. We ranked putative regulators by the sum of their outgoing importance scores and plotted the top 100 regulators per mouse in time by the response (Fig. 3a) to give an overview of the dynamics of regulators during ICB. This analysis confirmed different dynamic regulations between responders and non-responders, and therefore we proceeded to focus on the top 10 known transcription factors (TF) that formed central hubs in the network. For each of these TFs, we explored downstream

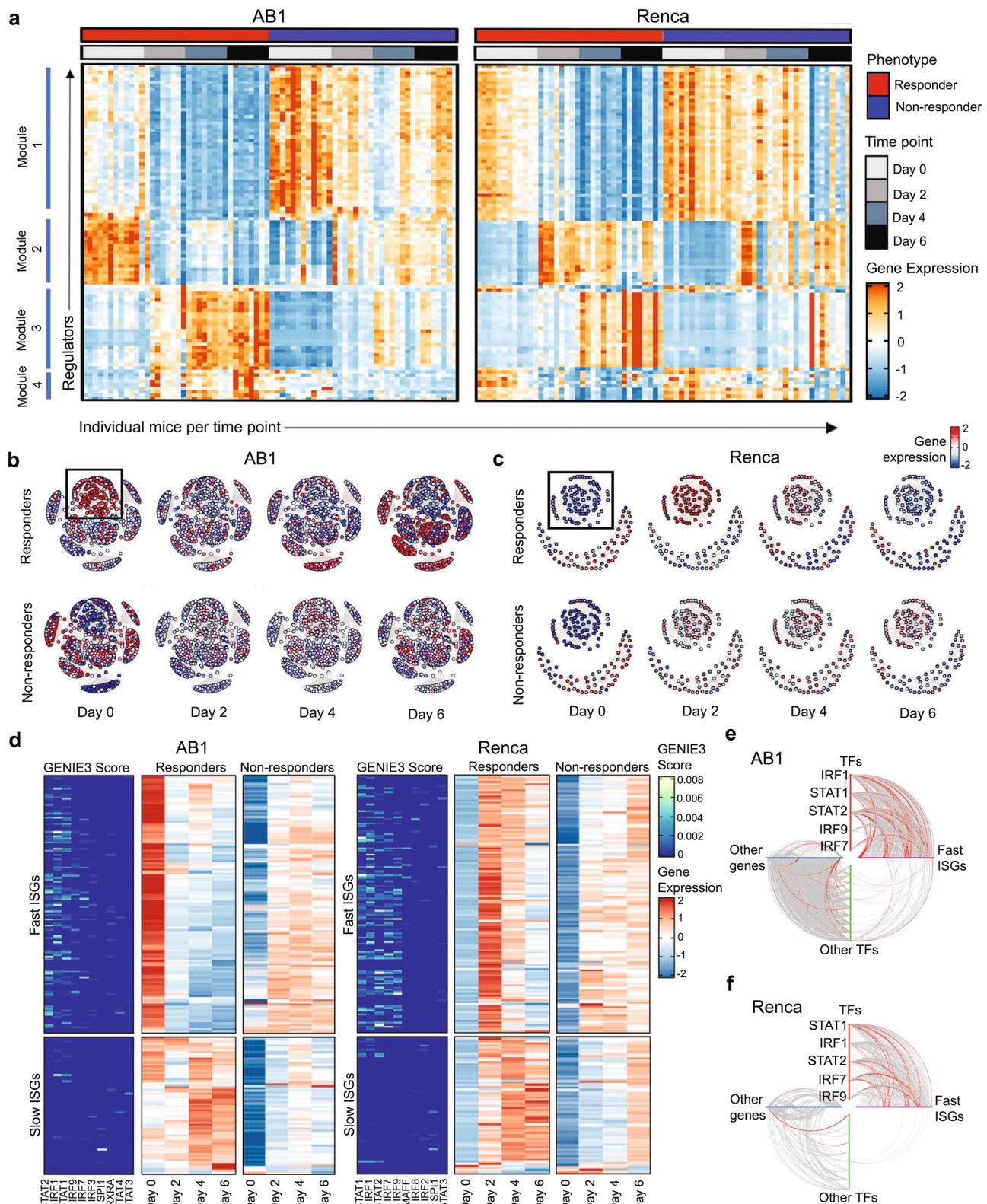

**Fig. 3 | Time-dependent changes in expression of a common subset of IFN regulators displaying on/fast-off kinetics distinguish ICB responders from non-responders. a** Expression of the top 100 regulators ranked by GENIE3 importance score plotted per mouse in time by the response, genes were grouped by hierarchal clustering. **b** GENIE3 subnetwork of direct interactions between TFs and their target genes in AB1 and (**c**) in Renca, separated into responders and non-responders, depicting gene expression over time, with on/fast-off IFN genes highlighted within the boxes. **d** Top 10 TFs with highest GENIE3 scores between ISGs for AB1 and Renca responders (left sub-panel), with average expression for ISGs in responders (centre sub-panel) and non-responders (right sub-panel) across time points. Hive plots of direct networks for AB1 (**e**) and Renca (**f**) with TF-to-ISG edges situated in the right upper quadrant. The top 10% of (high valued) edges by GENIE3 score are highlighted in red.

genes that were differentially expressed between responders and non-responders, and that contained a binding site for the TF within their promoter (Supplementary Fig. 4b). When we visualized the most connected TFs and their first order neighbours in these networks, we identified multiple gene modules with increased expression early during treatment (Fig. 3b, c). A module containing IFN-related genes exhibiting an on/fast-off dynamic response over time was found in both AB1 and Renca responders (Fig. 3b, c). This suggested that dynamic regulation of the IFN pathway is a determinant of ICB response, with a fast reduction in expression levels over the time course of the experiment.

Next, we investigated candidate upstream regulators of the IFN module. Because functional annotations of this module suggested that both type I and type II IFN contributed to the responder phenotype, we obtained a list of interferon-stimulated genes (ISGs) for both type I and type II IFN from the Molecular Signatures Database hallmark gene sets and isolated a "fast-on/off" subset of genes which showed on/off dynamics in responders. We analysed direct edges from TFs to these ISGs weighted by GENIE3 importance scores[28], which demonstrated that both AB1 and Renca showed similar on/fast-off kinetics of IFN signalling in responders, with on/fast-off changes by day 2 in AB1 and day 4 in Renca (Fig. 3d). In contrast, non-responders displayed a slower and less intense activation of ISGs which remained chronically active over time (Supplementary Fig. 4f). We confirmed that ISG expression segregated into an early and late phase, with the on/fast-off component containing genes stimulated by IFNγ, IFN α/β or a combination of both (Supplementary Fig. 4g).

For both models, GENIE3 scores indicated that these ISGs were regulated by a common set of TFs, namely IRF1, STAT1, STAT2, IRF7 and IRF9, (Fig. 3d), independent of tumour type. When the strength of these regulatory interactions are compared to all other interactions using hive plots[29], these TF to ISG connections dominate (Fig. 3e, f) in both models, reinforcing the essential contribution of these IFN-associated TFs to the dynamic response. Taken together, these results show that on/fast-off dynamics in IFN signalling are associated with response to ICB, driven by common transcription factors across different tumour models.

**Dynamic on/fast-off targeting of IFNβ improves response to ICB**
Because there is a large overlap in ISGs that are induced by type I or type II IFNs, we were unable to resolve which IFN type was driving the response based on the transcriptomic data alone and therefore interrogated this experimentally. To phenocopy the active IFN signature in vivo, prior to ICB, we pre-treated mice with intra-tumoural injections of poly(I:C), which is known to induce both type I and II IFNs, particularly IFNβ (Supplementary Fig. 5a, b)[30]. To mimic the subsequent on/fast-off-IFN signature, ICB was followed three days later by functionally blocking either type I or type II IFN signalling, using antibodies against the IFNα/β receptor (IFNAR1), IFNγ, or both (Fig. 4a). These studies were first done in AE17 mesothelioma, which is relatively resistant to ICB[17]. Pre-treatment with poly(I:C) improved the response rate to ICB (0% vs. 26.7% complete responders), which was significantly enhanced by the subsequent blockade of type I IFN (53.3% complete response, $p = 0.034$, Logrank test), but not type II IFN (Fig. 4b). We confirmed these findings in the AB1 mesothelioma model (33.3% vs 53.3% complete responders, Supplementary Fig. 5c). In both models, the beneficial effect of blocking type I IFN after administration was negated by blocking type II IFN simultaneously, which is in line with reports in patients with tumour defects in IFNγ signalling associated with acquired resistance to ICB[31]. These results demonstrate not only that type I IFN was responsible for the observed on/fast-off dynamics of IFN signalling in responders, but that these dynamics indeed mediated the therapeutic response. To explore the biological relevance of these IFN dynamics, we assessed the effect of blocking IFNAR1 before rather than after ICB initiation, or concomitantly with poly(I:C) prior to ICB. This

treatment completely abrogated both the response to ICB and the priming effect of poly(I:C). However if IFNAR was blocked after treatment with poly(I:C) alone, there was no detriment to the anti-tumour response, confirming the crucial time-dependent nature of IFN signalling underlying the therapeutic response to ICB (Supplementary Fig. 5d–g).

To further dissect these on/fast-off type I IFN dynamics, we first looked for signatures to computationally deduce the IFN subtype present in the responder tumour microenvironment. We used single-cell RNAseq data from cells stimulated with a diverse array of cytokines, including IFNβ[32], to construct a reference matrix using CIBERSORT[33]. We used this reference matrix to perform a deconvolution analysis on our bulk RNAseq data, which demonstrated that genes associated with IFNβ signalling, but not other cytokines, followed the on/fast-off IFN pattern in responders in both AB1 and Renca tumour models, suggesting IFNβ rather than IFNγ was responsible for these observed dynamics (Fig. 4c, d and Supplementary Fig. 6a, b). To confirm this, we treated AE17 tumour-bearing mice 3 days after administration of ICB with antibodies against either IFNβ, IFNα (subtypes A, 1, 4, 5, 11, and 13) or their shared receptor IFNAR1[34]. Mice treated with the antibody against IFNβ had a similarly increased response rate following ICB as the mice treated with the anti-IFNAR1 antibody (60% and 55.6% complete response vs. 10%), while mice treated with anti-IFNα displayed no increase in response versus controls (Fig. 4e). We repeated these experiments in the Renca model, which exhibited the same benefit of blocking IFNAR1 or IFNβ (22.2% and 40% complete response vs 0%), but not IFNγ or IFNα (Fig. 4f). We conclude that the beneficial effect of on/fast-off kinetics in type I IFN signalling after ICB is entirely dependent on switching off IFNβ.

As intra-tumoural administration of poly(I:C) is not approved for use in clinical practice in combination with ICB, we tested whether similar results could be achieved in the absence of initial IFN induction by poly(I:C). Time staggered blockade of type I IFN again improved ICB efficacy in AB1-bearing mice, which was dependent on IFNβ (Fig. 4g and Supplementary Fig. 6c), and we confirmed these findings in the AE17 and Renca models (Supplementary Fig. 6d, e). Although the response could be increased by blocking IFNβ after ICB, priming with poly(I:C) first to mimic the "on" IFN signature gave optimal results. Having established that blocking IFNAR1 abrogated the effect of poly(I:C) (Supplementary Fig. 5f), we investigated whether IFNα or IFNβ activity was driving the "on" signal in responders using recombinant cytokines. We found that priming with IFNβ, but not IFNα, increased the response to ICB ($p = 0.012$, Logrank test), and mimicking the on/off IFN signature targeting IFNβ was superior to targeting IFNα ($p = 0.027$, Logrank test) (Fig. 4h). Notably, the temporal aspect of scheduling the respective treatments was crucial, as treatment with anti-IFNβ concomitantly with ICB did not offer any therapeutic benefit, in contrast to administration 3 days after the first dose of ICB (Fig. 4i). These results confirm that temporal restriction of IFNβ activation underlies response to ICB.

To further define how these early kinetics of IFNβ modulates the T cell response later on during the therapeutic response, we dosed tumours with either recombinant murine IFNβ for 6 days (chronic) or for 3 days followed by an anti-IFNβ antibody (on/off), and analysed tumours by flow cytometry. On/off IFNβ signalling resulted in enhanced recruitment of CD8+ and CD4+ T cells (Fig. 4j, l), which were highly proliferative as defined by Ki67 expression Fig. 4k, m and Supplementary Fig. 7). While having little effect on the expression of checkpoints TIM3 and LAG3, on/off IFNβ activity upregulated PD-1 expression on CD8+ T cells, suggesting these are activated but not terminally exhausted T cells[35]. Chronic IFNβ did not cause upregulation of any immune checkpoints (Fig. 4k, m and Supplementary Fig. 7). We did not observe any effect on Tregs (Fig. 4n), but cannot fully exclude their involvement based on the literature regarding anti-

 

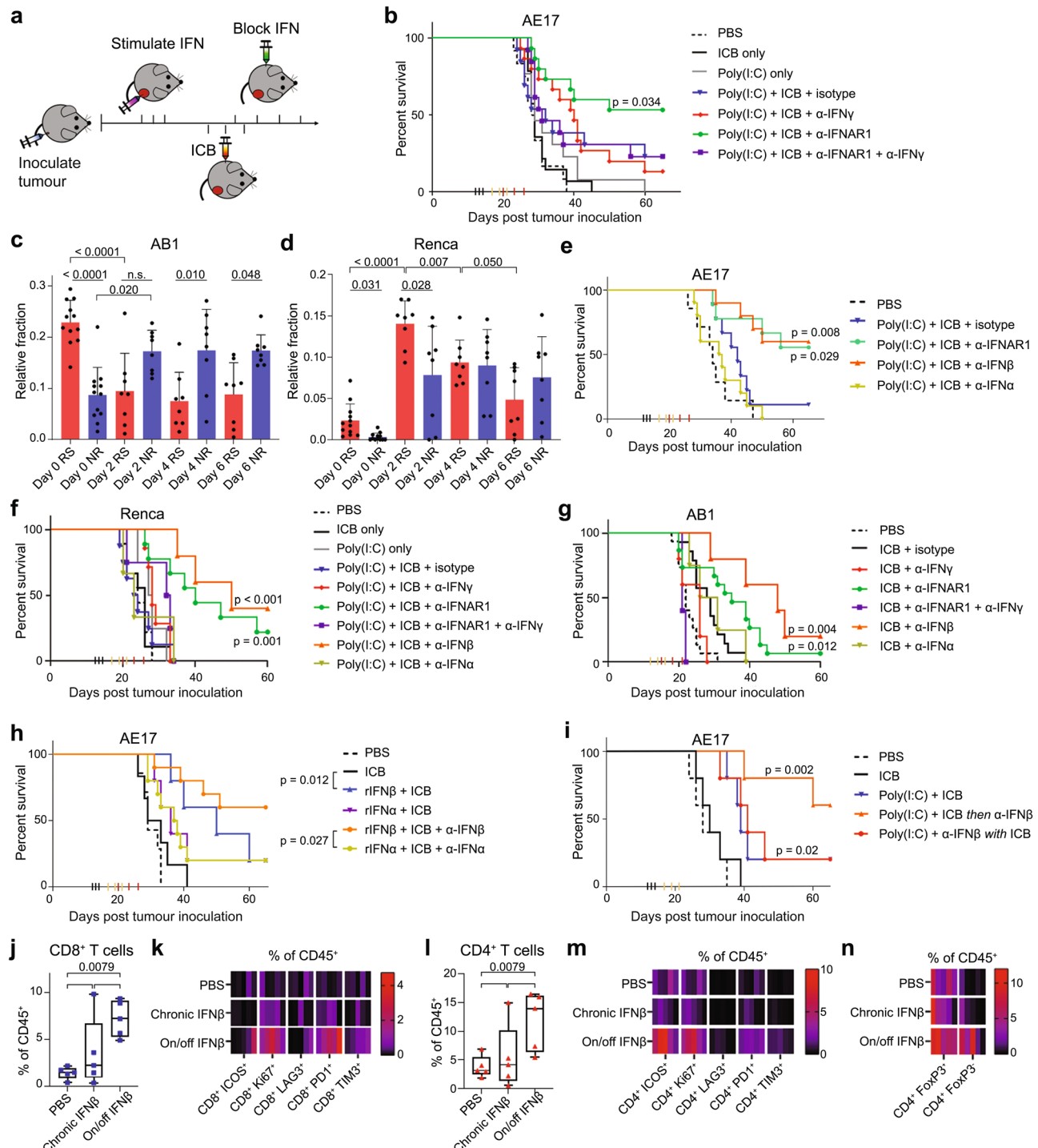

CTLA4 in mice[36]. Together, these data suggest that the dynamics of IFNβ signalling early-on during ICB affect T cell recruitment and activation.

### Inflammatory monocytes are the primary source of IFNβ in ICB responders

To further understand where the on/fast-off IFN signal was derived from, we performed single-cell transcriptome sequencing 1 h prior to ICB (Fig. 5a). We interrogated responder and non-responder samples using gene set enrichment analysis (Fig. 5b). We identified that particularly monocytes displayed elevated type I IFN signalling in responders (Fig. 5b). In particular, a specific monocyte sub-population drove the on/fast-off-IFN gene signature found in our bulk RNAseq data

(Fig. 5b, c). These monocytes (cluster 1, Fig. 5d) displayed high *Ly6c* expression (Fig. 5e), consistent with an inflammatory monocytic phenotype[37]. Repeating gene regulatory network inference on this single cell data confirmed that IRF1, IRF7, STAT1, IRF9 and STAT2 were major transcription factors driving the response in these cells (Fig. 5f, Supplementary Fig. 8a), supporting our network analysis results in bulk samples. In addition, we confirmed CD11b+/Ly6C[hi] monocytes attributed the highest expression of *Irf1*, a key ISG regulator, in tumour samples by flow cytometry (Fig. 5g).

We performed RNA velocity analysis and observed a trajectory from cluster 1 to cluster 2 monocytes characterized by a diminished activation of ISGs (Fig. 5h). Individual ISGs velocities along this trajectory are markedly diminished in non-responders compared to

**Fig. 4 | Targeting IFNβ in a directionally opposite, time-dependent manner improves the response to ICB. a** Treatment strategy. **b** Survival curves of AE17-bearing mice treated with poly(I:C), ICB followed by antibodies blocking type I and/or II IFN (*n* = 15 per group). **c, d** Deconvolution analysis on bulk RNA seq data from AB1 (**c**) and Renca (**d**) using an IFNβ-stimulated T cell signature between responders (red) and non-responders (blue). *N* = 8 to 12 biologically independent samples per group. Data presented as mean, bars represent standard deviation. *\*p ≤ 0.05,* *\*\*p ≤ 0.01, \*\*\*\*p < 0.0001* from two-way ANOVA with Tukey's multiple comparisons test. **e** Survival curves of AE17 bearing mice treated with poly(I:C), ICB followed by antibodies against the type I IFNs, IFNα or IFNβ compared to blocking their receptor IFNAR (*n* = 5 per group). **f** Survival curves of Renca-bearing mice treated with poly(I:C), ICB followed by antibodies against the type I and/or II IFN (*n* = 5 or 10 per group). **g** Survival curves of AB1 bearing mice treated with ICB followed by antibodies against type I and/or II IFN (*n* = 5 to 15 per group). **h** Survival curves of AE17-bearing mice treated with recombinant IFNα or IFNβ and ICB followed by antibodies blocking IFNα or IFNβ (*n* = 6 to 10 per group). **i** Survival curves of AE17-bearing mice treated with poly(I:C) and ICB, with antibodies blocking IFNβ given concurrently or 3 days after the first dose of ICB (*n* = 5 per group). **b**, **e–i**; *p* values from Logrank test, compared to control "Poly(I:C) + ICB + isotype" or "ICB + isotype". Dashes on x-axis represent day of dosing: black = poly(I:C) or rIFN, yellow = ICB, red = anti-IFN antibody. **j–n** Mice bearing AE17 tumours were given PBS, recombinant IFNβ i.t. daily for 6 days (chronic) or daily for 3 days followed by an anti-IFNβ antibody (on/off). Tumours were harvested and analysed by flow cytometry. **j** Proportion of CD8$^+$ T cells. **k** Proportion of CD8$^+$ T cells expressing immune checkpoints shown as % of CD45$^+$ cells. **l** Proportion of CD4$^+$ T cells. **m** Proportion of CD4$^+$ T cells expressing immune checkpoints shown as % of CD45$^+$ cells. **n** Proportion of Treg (FoxP3$^+$) or T-helper (FoxP3$^-$) CD4$^+$ T cells as % of CD45$^+$ cells. *N* = 5 biologically independent samples per group. **j** and **l** Box boundaries are the 25th and 75th percentiles, the horizontal line across the box is the median, and the whiskers indicate the minimum and maximum values. *P* values calculated from two-way ANOVA with Tukey's multiple comparisons test. Source data are provided in the Source Data file.

responders (Supplementary Fig. 8b). We, therefore, examined gene velocities for IFN-related TFs and ISGs in the on/fast-off gene signature. Hierarchical clustering showed separation of responder versus non-responder monocytes based on ISG velocities (Supplementary Fig. 8b). Along this trajectory, velocity analysis showed that monocytes down-regulated transcription of ISGs such as *Irf1*, and this was more pronounced in responders than non-responders (Supplementary Fig. 8c). Both these results are consistent with the on/fast-off IFN dynamic we observed in bulk RNAseq data (Supplementary Fig. 8d–f). In addition, we saw a change in expression from *Ly6c*$^{hi}$ to *Ly6c*$^{lo}$ monocytes along this trajectory (Supplementary Fig. 9a, b) and these changes at single cell level were compatible with *Ly6c* expression over time observed in the bulk RNAseq data (Supplementary Fig. 9c, d). As Ly6C1/2 is a marker of blood derived inflammatory monocytes[37], this suggests that these transcriptionally dynamic cells are likely infiltrating the tumour and differentiating. To further support this notion, we assessed the expression dynamics of the blood-derived monocytic marker CCR2[38], and found it followed the same kinetics as the *Ly6c* and on/fast-off gene expression patterns (Supplementary Fig. 9e, f).

To confirm whether monocytes were indeed the source of IFNβ in the tumour microenvironment, we used B6.129-Ifnb1$^{tmILky}$/J mice, which co-express IFNβ1 and eYFP from the *Ifnb1* locus, bearing AE17 tumours[39]. Tumours were analysed by flow cytometry, one day after intra-tumoural poly(I:C), which showed YFP-positive cells were all CD45$^+$, CD11b$^+$, MHC-II$^-$, F4/80$^-$, CD11c$^-$ with the majority expressing Ly6C, consistent with the single cell RNAseq results (Fig. 5i). To establish if these IFNβ cells were associated with response, we used the bilateral tumour model in B6.129-Ifnb1$^{tmILky}$/J mice, and analysed tumours 2 days after ICB by flow cytometry. This data confirmed, on the protein level, that responders had more IFNβ, which was expressed by CD11b$^+$Ly6C$^+$CCR2$^+$ cells (Fig. 5j). Together, these results pinpoint CD11b$^+$ monocytes as the key IFNβ producing cells in the tumour microenvironment.

**On/fast-off IFN signature in monocytic cells correlate with T-cell response in patients treated with ICB**

To validate our preclinical findings, we tested whether activation of IFN signalling also occurred in patients given ICB therapy, using single cell RNAseq data from a cohort of treatment naïve breast cancer patients, subsequently treated with anti-PD-1 and with paired T cell expansion data[25]. In this dataset, biopsies were taken prior to treatment with ICB and could be correlated with T cell expansion after treatment, a known indicator of clinical efficacy. In our analysis, we interrogated expander (responsive) and non-expander (non-responsive) samples pre-treatment using gene set enrichment analysis which confirmed a globally elevated on/fast-off IFN signature in expanders, which was highest in myeloid cells (Fig. 5k and Supplementary Fig. 10). Within the myeloid cell cluster, we found the *CCR2*$^+$ subpopulation in expanders

highly expressing the on/fast-off-IFN gene signature, consistent with our findings in the murine models of tumour-infiltrating inflammatory monocytes being the predominant source of type I IFN activity and IFNβ in particular (Fig. 5i, j, l and Supplementary Fig. 10). Furthermore, we demonstrate that our on/fast-off-IFN signature was superior in predicting response (Supplementary Fig. 10e), supporting the generalisability of our signature to human data.

## Discussion

Here, we report that on/fast-off activation of IFNβ in cancer is required for the therapeutic response to ICB, we dissect the contribution of IFNβ versus IFNα in the anti-tumour immune response, and we provide an example of time-dependent activation and inhibition of a drug target being required to achieve optimal anti-cancer effect.

Clinical studies have shown that an IFN gene signature is associated with treatment with ICB, and that an IFN-mediated signature can predict response[13,19,40,41]. However, the dynamic nature of an orchestrated innate and adaptive immune response against cancer cannot be adequately interpreted from a single snapshot[6,9]. Ideally, tumour biopsies would be frequently obtained early during ICB treatment to identify underlying mechanisms of response, but perhaps with the exception of some bone marrow cancers, this is usually not feasible due to anatomical tumour location and the requirement for invasive procedures. In addition, given the presence of intra-patient tumour heterogeneity, repeat biopsies may not provide a consistent representation of the tumour microenvironment[42]. Moreover, inter-patient heterogeneity makes it difficult to identify small yet meaningful biological differences underlying the response to ICB. Preclinical studies using cell line-derived tumours in syngeneic mice can negate some of these issues, including patient and tumour heterogeneity. Typically, comparisons are made between responsive and non-responsive tumour mouse models, or between untreated and treated mice[43], sometimes with suboptimal dosing as not to destroy the biological read-out of the perturbed tumour microenvironment[44]. We further refined this approach by comparing responders and non-responders to ICB within the same tumour model, sampling entire tumours over time[7]. Specifically, the bilateral models that we used are fully internally controlled; responders and non-responders are equally exposed to the relevant experimental variables, including potential inflammation associated with cancer cell inoculation, tumour size, surgical tumour removal (with sham surgery having no effect on symmetry[7,16]) and anaesthesia, in addition to the inherent genetic and environmental homogeneity of using inbred mice. Yet, mice display a stark dichotomy in response within the models. This difference in outcome is likely due to the stochastic nature of the induction of an effective anti-cancer immune response, which contains many switches, thresholds and feedforward and feedback loops[45]. Importantly, these models allowed us to identify dynamic immune response-intrinsic changes that govern

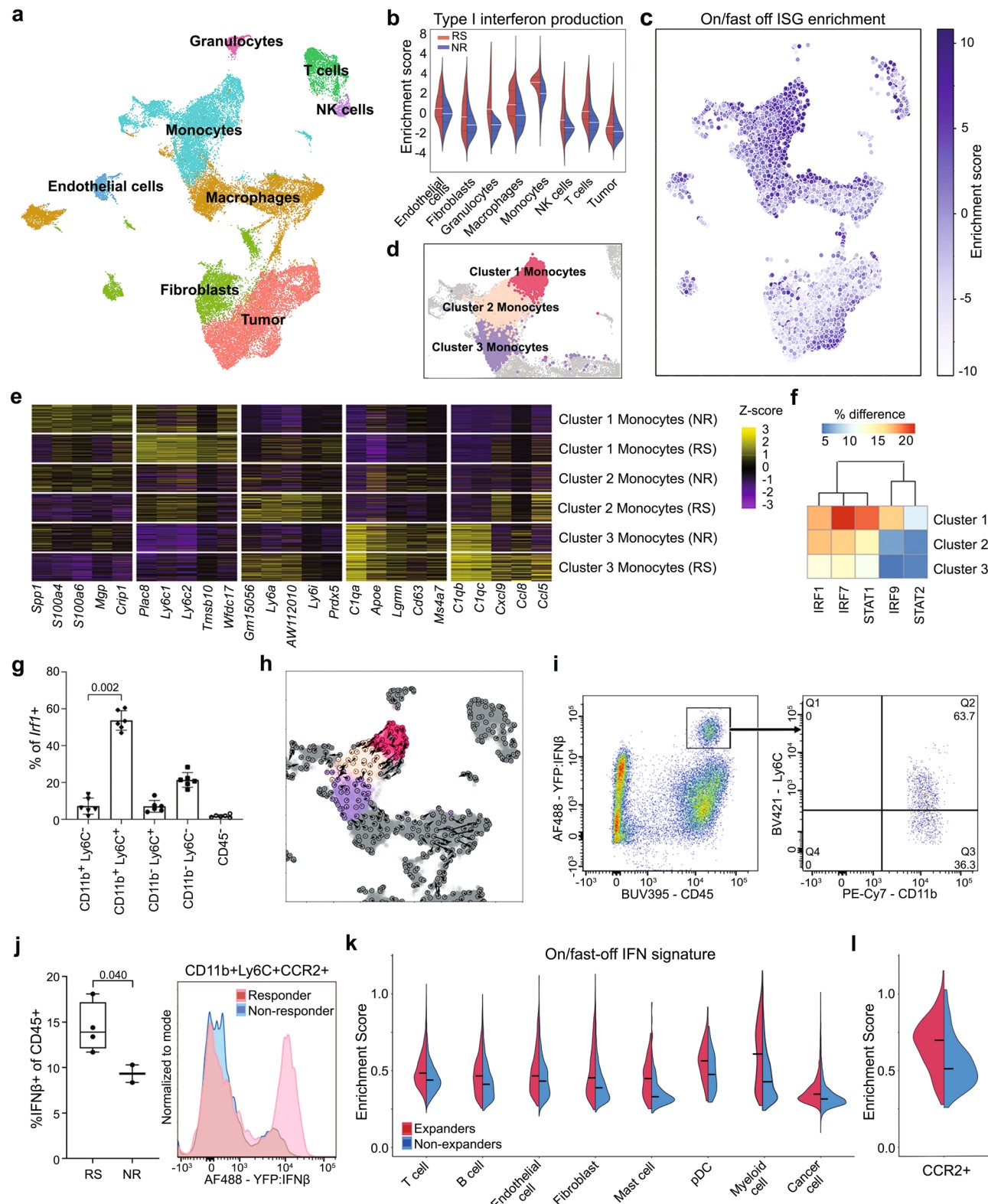

the therapeutic outcome in a highly homogenous genetic and environmental background.

Our results show that the on/fast-off dynamics of IFNβ signalling are crucial to the response to ICB, which can be therapeutically exploited using antibodies against IFNβ or its receptor IFNAR1, resulting in enhanced tumour clearing. Others have shown that secondary ICB-resistant cancers, which are cancers that initially responded but then relapsed, display chronically active IFN signalling[46,47]. This has resulted in the suggestion to co-treat patients with JAK inhibitors, which block both type I and II IFN signalling[46]. We extend these findings by demonstrating in intrinsically responsive tumours that type I IFN only, and more specifically IFNβ only has a dual function and that the response rate and depth of response can be improved by therapeutically mimicking these on/fast-off dynamics. We did not observe these effects when we blocked IFNα using an antibody that is specific for 5 subtypes[34]. We cannot exclude that other IFNα subtypes have an

**Fig. 5 | Single cell analysis identifies inflammatory monocytes as the primary source for IFNβ and on/fast-off type I IFN signalling in the responsive tumour microenvironment. a** UMAP visualization and annotation of cell subtypes from AB1 tumours prior to ICB (*n* = 6; 3 responders, 3 non-responders, 17935 cells). **b** Differential enrichment of type I interferon production in different cell subsets between responders and non-responders, the white line represents mean. **c** UMAP showing a gradient of diminishing on/fast-off ISG enrichment located in the monocyte cluster. **d** Identification of 3 distinct monocyte sub-clusters, highlighted on the UMAP plot, with conserved markers **e** for each cluster, separated by response. **f** IFN-related TF activation across AB1 monocyte clusters. **g** Flow cytometry of *Irf1* expressing cell populations (*n* = 6, *p*-value from two-sided Mann–Whitney U test, data presented as mean with error bars represent standard deviation). **h** Velocity analysis across monocyte clusters projected onto the UMAP plot. Arrows denote transcriptional gradient from cluster 1 to cluster 2. **i** Flow cytometry of poly(I:C) treated AE17 tumours from Ifnb1^tmILky mice to identify the phenotype of IFNβ producing cells (representative sample). **j** Flow cytometry on AE17 tumours from Ifnb1^tmILky mice two days after ICB, with response defined using the bilateral tumour model (*n* = 4 responders, 2 non-responders), to show the difference in IFNβ production from CD11b⁺Ly6C⁺CCR2⁺ cells. *p*-value from unpaired *t*-test with Welch's correction. Box boundaries are the 25th and 75th percentiles, the horizontal line across the box is the median, and the whiskers indicate the minimum and maximum values. Source data are provided in the Source Data file. **k** Enrichment of the on/fast-off signature in different cell subsets from patient breast carcinoma pre-treatment biopsies, comparing patients with and without T cell expansion after PD-1 therapy, black line represents mean. (*n* = 9 expanders, 20 non-expanders, 75790 cells). **l** Enrichment of the on/fast-off signature in a subset of patient tumour-derived myeloid cells (*CCR2*⁺), comparing expanders and non-expanders pre-treatment (*n* = 9 expanders, 20 non-expanders).

equally inhibitory effect on the therapeutic response. In contrast to previous studies that showed that chronically active IFN signalling causes relapse following ICB treatment[46,47], our data do not indicate that chronic IFNβ causes resistance, as such, but rather that a robust amplitude and short on/off timing are required for the initial therapeutic response.

Validation of these preclinical findings in patient samples is difficult, given the aforementioned issues with obtaining repeated samples in patients. However, using our bilateral mouse models, a previously identified gene expression signature predictive for response[16] has been extensively and independently validated by others by directly comparing the mouse data with multiple patient cohorts across different tumour types and different ICB antibodies[19–21]. In addition, we confirmed the presence of known indicators of clinical efficacy following ICB, such as increased CD8 T cell infiltration[24]. Lastly, several other groups have now independently demonstrated that bilateral models identify important and translationally relevant biology in the context of ICB[18,48]. We identified the requirement for on/off IFNβ activity in the context of ICB in three mouse models, across two strains, but further future validation in other tumour models is warranted.

In order to further validate our preclinical findings, we used single cell RNAseq data from a cohort of breast cancer patients treated with anti-PD-1[25], which confirmed that inflammatory tumour-infiltrating monocytes were mainly responsible for the on/fast-off IFN signal in these patients, and that enrichment of these cells was correlated with T cell expansion. Although these patient samples do not provide the temporal granularity of four-time points or definitive clinical response data, like our murine study does, which limits the ability to identify kinetic mechanisms underlying ICB efficacy, they do highlight the contribution of monocytes in the response to ICB. Whether the down-regulation of IFNβ is involved in driving the T cell response, or whether chronic IFNβ signalling provides cancer cells with a survival advantage under immune pressure remains to be established. Interestingly, in that context, it has been reported that cancer cells that are unable to downregulate the IFNAR1 receptor during melanoma development respond better to ICB compared to normal melanoma cells[49]. Although the validation of the results in clinical samples provides confidence that the therapeutic targeting of IFNβ in a time-dependent manner could improve the response to ICB in patients, future clinical trials are needed to assess IFNβ as a biomarker and dynamic drug target to define the optimal time to start blocking after ICB.

The finding that time-dependent aspects of type I IFN signalling contribute to a powerful anti-tumour immune response resonates with findings from viral immunology, where type I IFN is required for acute clearance of viral infections such as hepatitis C, and it is even used therapeutically in that context[50]. Yet, paradoxically, blocking IFNAR1 can be beneficial for the control of chronic viral infections, as has been shown for LCMV or coronavirus infections[34,51–53]. We propose that the anti-tumour response following ICB mimics aspects of the acute or chronic anti-viral immune response, resulting in either swift regression or non-response, respectively. This notion is also in line with recent findings showing inflammatory monocytes as major responders to type I IFN production in viral immunity, as well as our results and those from others in the context of ICB in cancer[54,55].

The interaction between the immune system and cancer cells is often conceptualized as a cycle, which can be pushed at any level, at any given time, to induce the appropriate momentum[56]. Instead, our data suggest a continuous changing landscape of the immune response, where interventions have a time-dependent effect, even to the point that the exact same target must be modified in a diametrically opposite manner; by providing excess recombinant IFNβ first, followed by blocking antibodies against IFNβ later. In oncological treatments in general, drugs, including in combination, are typically administered empirically until they are no longer effective or toxicities preclude continuation. Our results challenge this approach and could have important implications for drug discovery research, demonstrating that in order to obtain optimal clinical effect, some targets need to be therapeutically modulated in a time-dependent, bidirectional manner. As recombinant IFNβ has been FDA approved for multiple sclerosis and antibodies targeting the IFNβ/IFNAR1 pathway have been fully developed in the context of autoimmunity[57], these results can be readily translated into the clinic.

## Methods

### Statement

All animal experiments described in this study were performed according to animal protocols approved by the Harry Perkins Institute for Medical Research animal ethics committee. This article does not contain any studies with human subjects performed by any of the authors.

### Mice

BALB/cArc, Balb/cAusB, C57BL6/J or Ifnb1^tmILky/J mice (C57BL6/J background) 8–12 weeks of age were used for all experiments. BALB/cArc or C57BL6/J mice were obtained from the Animal Resource Centre (Murdoch, WA), Balb/cAusB mice were obtained from the Harry Perkins Institute for Medical Research Bioresources Centre South (Murdoch, WA). Ifnb1^tmILky/J mice, generated by knock-in of a yellow fluorescent protein (YFP) reporter cassette into the endogenous Ifnb locus[39], were imported from The Jackson Laboratory (Bar Harbour, Maine) and maintained at the Harry Perkins Institute for Medical Research Bioresources Centre South (Murdoch, WA). All mice were housed at the Harry Perkins Institute of Medical Research Bioresources Facility North under specific pathogen-free conditions. Mice were fed Rat and Mouse cubes (Specialty Feeds, Glen Forrest, Australia) and had access to water ad libitum. Cages (Techniplast, Italy) were individually ventilated with filtered air, contained aspen chips bedding (Tapvei, Estonia) and were supplemented with tissues, cardboard rolls and wood blocks as environmental enrichment, and were changed every 14 days. Mice were housed at 21–22 °C, 60% humidity with 12 h light/

dark cycle (06:00–18:00). Sentinel mice ($n = 3$) in the animal facility were screened monthly for a standard panel of bacteria and fungi, ectoparasites, endoparasites, non-pathogenic protozoa and viruses (Cerberus Sciences, Australia). All experiments were conducted in compliance with the institutional guidelines provided by the Harry Perkins Institute for Medical Research animal ethics committee (approval numbers AE047, AE091, AE157, AE201).

## Cell culture

Cell lines AB1 and AE17 were obtained from CellBank Australia. Cell line Renca was kindly donated by Dr. E. Sotomayor and Dr. F. Cheng (University of South Florida, Tampa, FL). Cell lines were maintained in RPMI 1640 supplemented with 20 mM HEPES, 0.05 mM 2-mercaptoethanol, 100 units/ml penicillin/streptomycin (Thermo Fisher), and 10% FCS (Invitrogen, Mulgrave, Australia). Cells were grown to 70–80% before passage and passaged 3–5 times before inoculation. Cells were frequently tested for mycoplasma by PCR and remained negative. Cell lines were validated yearly by flow cytometry for MHC class I molecules H2-Kb (consistent with C57BL/6) and H2-Kd (consistent with BALB/c), and for fibroblast markers E-cad, EpCam and PDGFRα (negative) and by PCR for mesothelin (positive for AB1, negative for Renca).

## In vivo treatments

When cell lines were 70–80% confluent, they were harvested and washed 3 times in PBS. $5 \times 10^5$ cells in 100 µl were inoculated subcutaneously (s.c.) onto the lower right-hand side (RHS) flank (for single inoculations) or both flanks (for dual tumour inoculations) using a single 26 G needle per injection. Mice were randomized when tumours became palpable, approximately 3–5 days after tumour inoculation.

## Surgery experiments

A detailed protocol for the surgery experiments has been previously described[17]. For the pre-treatment samples, tumours were resected eight (AB1) or 10 (Renca) days post tumour inoculation, when tumours were ~9 mm², and mice were administered ICB 1 h after surgery. For the post-treatment samples, tumours were resected 2, 4 or 6 days after the first administration of ICB. Mice were dosed with 0.1 mg/kg buprenorphine in 100 µl s.c. (30 min prior) and anesthetized using isoflurane (4% in 100% oxygen at a flow rate of 2 L/min). Whole tumours and the corresponding draining inguinal lymph node on the right-hand side were removed by surgical excision and immediately immersed in RNAlater (Life Technologies, Australia). The wound was closed with staples (Able Scientific, Australia). Mice were placed in a heat box for recovery. The remaining tumour was monitored for the response as an indicator of response for the removed tumour. Mice were designated as responders when their tumour completely regressed, and they remained tumour free for up to 4 weeks after treatment. Mice were designated as non-responders if their tumours grew to 100 mm² within 4 weeks after start of treatment, similar to saline-treated controls (Supplementary Fig. 1). Mice that had a delay in tumour growth or partial regression were designated as intermediate responders and excluded from the analysis. For internal consistency, we only used experiments in which mice displayed a dichotomous response, i.e. in any cage, there had to be at least one non-responder amongst responders or vice versa.

## In vivo ICB treatment

The anti-PD-L1 hybridoma (clone MIH5) and the anti-CLTA4 hybridoma (clone 9H10) were cultured in IMDM containing 1% of FCS and gentamycin at Bioceros (Utrecht, The Netherlands). Clarified supernatants were used to purify the antibody using affinity chromatography. The antibodies were sterile and formulated in PBS. Alternatively, antibodies from the same clones were obtained from BioXcell (New Hampshire, US). Mice received an intraperitoneal (i.p.) dose of 100 µg of anti-CTLA4 and 100 µg anti-PD-L1 combined in 100 µl phosphate-buffered solution (PBS). Mice received additional doses of 100 µg anti-PD-L1 two and four days later. In previous experiments[58], no difference in the effect of control IgG versus PBS was found, and therefore vehicle controls received PBS alone.

## Tumour preparation for RNA sequencing

Whole tumours and lymph nodes were surgically resected, the surrounding tissue was removed and immediately submerged in RNAlater (Life Technologies, Australia). Samples were stored at 4 °C for 24 h, after which supernatant was removed and samples transferred to −80 °C. Frozen tumours were dissociated in Trizol (Life Technologies, Australia) using a TissueRuptor (QIAgen, Australia). RNA was extracted using chloroform and purified on RNeasy MinElute columns (QIAgen, Australia). RNA integrity was confirmed on the Bioanalyzer (Agilent Technologies, USA). Library preparation and sequencing (50 bp, single-end) was performed by Australian Genome Research Facility, using Illumina HiSeq standard protocols.

## Alignment and differential expression

We processed a total of 144 RNA-seq single-end read samples across four-time points in two mouse models. After reviewing quality control on all samples using FastQC software, we used Kallisto[59] (v0.43.0) for transcript abundance estimation. Following alignment, we performed differential expression analysis with Sleuth (v0.29.0)[60]. We compared responders and non-responders using a model containing time-point and response as covariates using a likelihood ratio test. We aggregated $p$-values from transcript differential expression to gene-level results with transcript-to-gene mapping relying on the latest Gencode reference M25 (GRCm38.p6) using Lancaster's method[61]. Genes were deemed differentially expressed at a false-discovery rate of less than 5%, regardless of fold change (Supplementary Data 1).

## RNAseq analysis of dynamic gene expression data

We used a deconvolution approach to deduce the cell subtypes present in the responder and non-responder tumour microenvironment at each time point. The CIBERSORTx[33] algorithm was used to estimate the relative proportions of 22 mouse hematopoietic immune cell types based on the transcriptomic profiles of each sample, using the LM22 matrix as a reference. We broadly classified the 22 cell sub-types into 9 major populations by collapsing several related subpopulations as follows: B cells include memory, naïve, and plasma cells; CD4 T cells include CD4 memory resting, memory activated, naïve, follicular helper; macrophages include M0, M1, and M2 phenotypes; NK cells include activated and resting cells; dendritic cells include activated and resting cells; granulocytes include neutrophils, mast cells resting, mast cells activated and eosinophils. Before analysis, transcript-level data were library-sized, and gene length normalized to TPM. Raw data for all subpopulations is available in the Source Data File. Comparison of CD8⁺ T cell score in responders and non-responders for each time-point was done by one-way ANOVA with Benjamini–hochberg correction for multiple comparisons (Supplementary Data 2).

We clustered time course RNAseq data using the fuzzy c-means (FCM) clustering algorithm Mfuzz[62] in the TCseq package[26]. Z-normalised/scaled counts were used in the algorithm and expression profiles were grouped clusters ($k = 6$) based on their dynamic patterns. We used a Pearson correlation score on trend lines to compare trends of each cluster between responders and non-responders, to identify which patterns were unique to responders. Each matching cluster between the two models had overlapping genes extracted (Supplementary Data 3) and enrichment of per-cluster genes was performed using Enrichr[63].

To acertain transcription factors involved in these processes, we constructed two networks, one for Renca responders and one for AB1 responders. We used the GENIE3 algorithm[27], which achieved the best performance on the DREAM5 network inference challenge[64]. To construct each gene regulatory network, we used 36 responder samples

across all time points as input to the GENIE3 algorithm. Since GENIE3 requires gene counts as input, we summarized transcript abundances derived from Kallisto as gene counts using the Bioconductor tximport[65] package. We ran GENIE3 (v1.8.0) with default parameters (treeMethod = "RF", k = "sqrt", nTrees = 1000).

We pruned GENIE3 output to isolate biologically relevant edges, incorporating DE information and transcription factor binding site predictions from JASPAR[66] (Supplementary Fig. 4). We identified and retained "direct connections", defined as connections between a transcription factor (TF) to differentially expressed genes only if the TFBS for the TF was situated in a genomic window 400 base pairs upstream or 300 base pairs downstream of a DE gene's transcription start site (TSS). TSS sites for differentially expressed genes were obtained from the UCSC genome browser[67]. The BEDtools[68] "window" function was used to obtain all direct regulatory interactions and GENIE3 importance scores appended to these interactions using the R data.table package (v1.12.2). We plotted the top 100 regulators ranked by GENIE3 importance score per mouse in time by response. Genes were grouped by k-means clustering, resulting in 4 main modules based on based on expression profiles across time (Supplementary Data 4).

### Network visualization and analysis of IFN-stimulated genes

We visualized the induced subgraph of the top ten nodes and their first-order neighbours in both AB1 and Renca using the R igraph package with the Kamada-Kawai[69] layout. We assigned colour to genes in the network by their average gene expression across time, normalized by Z-score. From inspection of these graph visualizations, we confirmed in both AB1 and Renca that these subnetworks were enriched for interferon-related transcription factors (Irf1, Stat1, Stat2, Irf7, Irf9) and their direct downstream targets. Pathway analysis on these hubs using Enrichr showed statistically significant enrichment of terms relating to both Type I and Type II interferon signalling, suggesting that dynamic changes in these two pathways were crucial to the checkpoint blockade response.

To identify ISGs involved in Type I and Type II interferon signalling in these direct networks, we first constructed gene list from gene sets for interferon alpha/beta signalling and interferon gamma signalling from the Molecular Signature Database (MSigDB)[28,70] converted into their mouse equivalents using BioMart[71]. After conversion, we retained 88/97 genes from the alpha dataset and 186/200 genes from the gamma dataset (Supplementary Fig. 4).

To obtain the fast-on/off signature used in our analysis, we refined the gene list described above by leveraging expression data for AB1 and Renca data. Specifically, we performed K-means clustering on these genes across the 4 experimental time points, isolating a "fast-on/off" subset of genes which showed upregulation in Day 0 in AB1 and Day 2 in Renca responders. The complete on/fast-off IFN gene signature mapped to human orthologs is given in Supplementary Data 5.

We confirmed, based on GENIE3 scores, that regulation of these on/fast-off genes was confined to just 5 key interferon-related transcription factors - Irf1, Stat1, Stat2, Irf7 and Irf9, with minimal regulatory impact from other transcription factors in these networks (Supplementary Fig. 4 and Fig. 3). To visualize TF-to-ISG GENIE3 scores and expression profiles for ISGs common to both AB1 and Renca networks, we used the R ComplexHeatmap[72] package. GENIE3 scores were extracted from each direct regulatory network, setting any non-existent interactions in the matrix to zero prior to visualization.

### Hive plot generation

To overcome visual biases from traditional network layouts, we used a hive plot visualization[29]. For each network, we used a 4-axis hive plot in the HiveR package, allowing us to partition edges to visualize interferon-related signalling in our networks. The following axes were used, colour coded in the following way (Fig. 3d, e): Red axis –

interferon related TFs (IRF1, STAT2, STAT1, IRF7 and IRF9); Green axis – non-interferon related important TFs, comprising the union set of top TFs from both networks in Supplementary Fig. 4; Purple – Downstream gene targets in the "on/fast-off ISG set"; Blue – Other genes in the direct networks not in 1, 2 or 3. For axis 3, the fast-ISG set was derived from k-means (k = 2) clustering on time course expression data from the AB1 responder data set. This node topology allows us to more easily visualize the "quadrant" of graph edges from IFN-related TFs to on/fast-off-ISGs, demonstrating that this quadrant contained edges with high value GENIE3 scores (above 0.9 quantile) denoting important dynamic regulatory links in the network.

### Single cell sample pre-processing

AB1 and renca tumours were surgically removed 1 h prior to ICB administration and immediately submerged in cold PBS, cut into 1–2 mm pieces with a scalpel blade and dissociated using the Gentle-MACS system (Miltenyi). Cell suspensions were frozen in RPMI medium containing 50% FCS and 10% DMSO. Cryopreserved single cell suspensions were rapidly thawed in a 37 °C water bath and prepared for single cell library construction as previously described[73]. Libraries were constructed using the 10X Chromium 3' workflow (version 2 chemistry) as per the manufacturer's directions. We aimed to capture 9000 cells per sample. Libraries were quantified using the TapeStation D1000 kit (Agilent). Sequencing was performed by Novogene, using NovaSeq S2 flowcell sequencing protocols.

For single cell analysis, we processed FASTQ files from 6 AB1 and 6 Renca samples using cellranger v3.0 (10X genomics). For each sample, we performed demultiplexing and read alignment using the cellranger count function, using cellranger's pre-supplied mm10 reference with an expect-cells parameter of 6000.

### Clustering, visualization and cell annotation

We used the Seurat[74] (version 3.14) R package to combine samples for downstream analysis. Gene counts were normalized against both sequencing depth and also against the percentage of mitochondrial DNA in each cell using negative binomial regression. The resulting Pearson residuals from these processing steps were used for downstream PCA, cluster identification and UMAP embedding and visualization.

To avoid subjective biases in cell identification, we used an automated labelling strategy based on bulk RNAseq references. The R package SingleR[75] was used in "cluster mode" using species-specific annotation references provided with the package. For annotation of human single-cell data, we used the human primary cell atlas ref. [76]. For annotation of mouse data, we used the mouse RNAseq dataset from Benayoun et al.[77].

Clusters were defined from Seurat's FindClusters function at default (0.8) resolution. Similarly, labelled clusters were merged. We confirmed that this approach was robust to cluster size by showing that labels were consistent even when cluster size was modified by changing resolution parameter in the FindClusters function. We performed annotation diagnostics by checking cell cluster identities in our AB1 and Renca samples against the ImmGen[78] reference. We found both references to be in agreement. The distribution of cell types and monocyte clusters between responders and non-responders is provided in Supplementary Data 6 visualised using the scCODA package[79].

### Single cell differential expression analysis and conserved marker analysis

We performed differential expression using the FindMarkers function in Seurat. Genes were deemed differentially expressed at an absolute log-fold change of 0.5 and a q-value of below 0.05 using the non-parametric Wilcoxon test. In Renca, we observed far fewer differentially expressed genes between responders and non-responders at Day 0 across all cell types, consistent with our bulk RNAseq data.

## Label transfer and interspecies integration of single-cell data samples

We used Seurat's "label transfer" functions, allowing cell identities from a reference sample to be projected onto our target dataset to identify tumour cells and to compare monocytes from mouse and human single-cell data. To construct a reference sample of mesothelioma cells, we sequenced samples from AB1 mesothelioma tumours in which the tumour cells were tagged with influenza hemagglutinin[80]. The cellranger index was rebuilt to incorporate the tag sequence. After alignment, any cell containing the HA-tag was labelled as a tumour cell and used as a reference for label transfer. Clusters containing more than 10% tumour cells were deemed to be putative tumour clusters.

To label Renca tumour cells, we used a reference dataset of mouse kidney single cells[81] in which the authors identified gene markers mapping to anatomical elements of the mouse renal tubular system, since these Renca tumours recapitulate renal tubule elements. We selected clusters that expressed the highest average expression of markers specific proximal and distal tubules (*Lrp2, Slc27a2*). Cells in these reference clusters had their identities projected onto the Renca dataset.

## Copy number variation analysis in tumour cells

Tumour cells usually display evidence for somatic large-scale chromosomal copy number alterations, such as gains or deletions of entire chromosomes or large segments of chromosomes. We used the inferCNV R package[82] to check tumour cell identity after label transfer. In both AB1 and Renca, tumour clusters labelled by Seurat's projection strategy were in good agreement with clusters of cells deemed to be tumour clusters based on the existence of copy number changes inferred by inferCNV.

## Gene set enrichment analysis on single cell data

We used the SCDE/pagoda[83] package, which detects statistically significant coordinated variability at the single cell level. Briefly, from the original 38 K cells in our AB1 samples, we constructed KNN error models for the 17 K surviving default SCDE's library size filters. For gene sets, we tested GO terms for interferon production and interferon response extracted from the org.Mm.eg.db Bioconductor package and also our "custom" gene set of fast-ISGs derived from bulk expression data. We visualized enrichment scores using python's Seaborn scatter plot with a colour scale mapped to enrichment score intensity.

Single cell analysis on the validation dataset of human breast cancer patients[25] was conducted using the escape[84] R package (v1.3.1), which provides convenience functions for the GVSA algorithm[85]. Random subsampling (without replacement) was performed to create input batches of 20,000 cells to the GVSA algorithm, and the results were pooled for visualization with the R ggplot2 package[86]. Filtering, normalization and scaling were performed as outlined in the original paper[25]. Cell clusters were labelled using the labels assigned by the authors. The statistical significance of "global" pathway enrichment in both the mouse and human data and magnitude of cell-specific differential pathway enrichment (comparing responders vs. non-responders across cell types) is shown in Supplementary Data 7. Pagoda2 overdispersion analysis was used to compare enrichment scores of gene sets (Supplementary Data 7). We used the interferon gene signature from Bassez et al.[25]. We used pre-ranked GSEA to compare signature enrichment between responders and non-responders (Supplementary Data 7) using the product of log fold-change and *p*-value as a metric[87].

## SCENIC network analysis

We performed network inference to analyse our single-cell data using SCENIC[88]. We summarized the result of these analyses using a difference heatmap between responders and non-responders of average binarized TF activity per Seurat cluster. Specifically, the regulon binarization scores were averaged across clusters, separated by the response and the difference between responder and non-responder averages were visualized using the R pheatmap (v1.0.12). To visualize these results, we created a heatmap showing the difference in percentage of TF activation per cluster for the IRF-related TF (Irf1, Stat1, Stat2, Irf7 and Irf9) (Fig. 5f). The full heat map of differential transcription factor activation in AB1 monocytes is displayed in Supplementary Fig. 8.

## Velocity analysis

RNA velocity quantifies cell transcriptional activity by modelling the time derivative of gene expression states. We used the velocyto[89] package for this analysis. Per-sample loomfiles, containing quantified spliced vs. unspliced transcripts were combined using the python loompy (v3.0.6) package. After count normalization, filtering and feature selection on these genes, approximately 2.5 K genes survived these filtering steps to be used for gene velocity modelling.

We compared "transcriptional momentum" of AB1 monocytes, which we defined as the squared L2 norm for each cell's embedding vectors with respect to their UMAP[90] coordinates. We compared KDE distributions for momentum in various AB1 clusters, separated by the response (Supplementary Fig. 8). To compare Renca and AB1 monocytes, we repeated our velocity analysis on using an embedding of all 12 samples from both AB1 and Renca single-cell data. On this common embedding, momentum calculations show that AB1 responders have a higher transcriptional velocity than Renca monocytes in responders (Supplementary Fig. 8).

As an additional check that differences in transcriptional momentum were, in part, due to differences in interferon signalling, we examined gene velocities for IFN-related TFs and ISGs in the on/ fast-off component gene set which survived the above-mentioned filtering. We extracted the normalized velocities of 42 of these genes, which survived data preprocessing. Hierarchical clustering showed separation of responder versus non-responder monocytes based on ISG velocities (Supplementary Fig. 8).

## Cytokine stimulation estimation

We aimed to verify whether there was evidence of an IFNβ-induced signature, irrespective of the cell type involved. We used the only available dataset which compared known cell differentiation cytokines under co-stimulatory conditions including IFNβ stimulation. We used a deconvolution approach to deduce the IFN subtype present in the responder tumour microenvironment to share the weighting of genes conserved across the different stimulatory conditions. The CIBERSORT algorithm[31] was used to estimate the relative proportions of 7 cytokine-induced T cell signatures based on the transcriptomic profiles of each sample, where the induced T cell gene signature developed by Cano-Gamez et al.[32] was used as a reference (Supplementary Fig. 6). We broadly classified the 94 samples into 7 major populations by collapsing several related sub-populations by their cytokine treatment: IFNβ, Resting, Th17, Th2, Th1, Th0, iTreg to generate the reference file (Supplementary Data 8). Prior to analysis, gene count data for both AB1 and Renca was normalized to TPM. The data were filtered to retain genes with an TPM value >0.3 in at least 8 samples (being the smallest experimental group size). CIBERSORT was run on AB1 and Renca separately, with quantile normalization disabled as recommended for RNAseq data.

## IFN modulation drug dosing schedules

As we aimed to boost the IFN response using poly(I:C) in the tumour microenvironment prior to administration of ICB to improve response, the timing of administration of ICB antibodies was scheduled late, as to have a low background response rate to ICB. Dosing with drugs commenced on day 12. Poly(I:C) (HMW, Invivogen) was dosed intratumourally at 50 µg daily for 3 days. Recombinant murine IFNα or IFNβ (Biolegend) was dosed intratumourally at 40,000 U daily for 3 days. ICB dosing began 3 days after the final dose of poly(I:C) or

recombinant IFN, on day 17. For treatment arms which had ICB only, dosing began the same day as poly(I:C) or recombinant IFN. Anti-IFNAR1 (Bioxcell, clone MAR1-5A3, 0.5 mg i.p.), anti-IFNγ (Bioxcell, clone XMG1.2, 0.5 mg i.p.), anti-IFNα (Leinco, clone TIF-3C5, 1 mg i.p., which blocks subtypes A, 1, 4, 5, 11 and 13), anti-IFNβ (Leinco, clone HDß-4A7, 0.6 mg i.p.), or IgG2a isotype (Leinco, clone C1.18.4, 0.6 mg i.p.). Treatment began 3 days after the first day of ICB administration, on day 20, and dosed every 3rd day for a total of 3 doses. For treatment schedules without poly(I:C), treatment with ICB commenced on day 10 for AE17 or day 12 for AB1 and Renca, followed by anti-IFN treatment 3 days after. Treatments were administered by one investigator (RMZ), while tumours were measured at least 3 times weekly using calipers by another researcher (TC) who was blinded for treatment allocation, to guarantee blinded assessment of the primary endpoint.

### Flow cytometry of *Irf1*+ cells

For flow cytometric analysis of *Irf1* gene expression in different cell subsets, AB1 tumours ($n = 6$) were harvested 6 days after inoculation and immediately submerged in cold PBS, cut into 1–2 mm pieces with a scalpel blade and dissociated using the GentleMACS system (Miltenyi). Fc block (anti-CD16/CD32, BD) was used for 10 minutes on ice. Cells were stained with Fixable Viability Stain 780 (BD) for 30 minutes at RT, to discriminate live cells. Cells were stained using antibodies for surface markers for 30 minutes at 4 °C (Supplementary Table 1). To identify *Irf1*+ cells, we used the PrimeFlow Kit (cat # 88-18005-204, Invitrogen). Briefly, cells were fixed in RNA fixation buffer 1, permeabilized with RNA Permeabilization buffer with RNase inhibitors, then fixed with RNA fixation buffer 2 before using Target Probes against *Irf1*. The signal was then amplified, followed by addition of fluorescent label probes (Alexa Fluor 647). Data were acquired on a BD Fortessa flow cytometer and analysed using FlowJo software (TreeStar). Cells were gated on *Irf1*+, followed by CD45+ to identify immune infiltrating cells, and CD45− non-immune cells (e.g. tumour cells). Immune cell populations were analysed by their expression of CD11b and Ly6C: Ly6C− Monocytes (CD11b+, Ly6C−); Ly6C+ Monocytes (CD11b+, Ly6C+, also F4/80−, CD3−, CD335−); Other Ly6C+ cells (CD11b+/−, Ly6C+); and remaining cells (CD11b−, Ly6C−). See Supplementary Fig. 11 for gating strategies.

### Flow cytometry of YFP+ cells

For flow cytometric analysis to find which cells express IFNβ, we used B6.129-Ifnb1tm1Lky/J mice, which co-express IFNβ1 and eYFP from the *Ifnb1* locus[35]. AE17 tumours ($n = 3$) were treated with poly(I:C) i.t. 21 days after inoculation and harvested 24 h later. To compare responder and non-responder tumours, Ifnb1tm1Lky/J mice were given bilateral AE17 tumours, treated with anti-CTLA4 and anti-PD-L1, and one tumour was surgically removed 2 days after initiation of therapy. Tumours were cut into 1–2 mm pieces with a scalpel blade and dissociated using the GentleMACS system (Miltenyi). Fc block (anti-CD16/CD32, BD) was used for 10 minutes on ice. Cells were stained with Fixable Viability Stain 780 (BD) for 30 minutes at RT, to discriminate live cells. Cells were stained using antibodies for surface markers for 30 minutes at 4 °C (Supplementary Table 1). To detect YFP+ cells, cells were fixed using a cytofix/cytoperm kit (cat # 554714, BD), then stained using antibodies against GFP (YFP cross-reactive) in perm buffer overnight. Data were acquired on a BD Fortessa flow cytometer and analysed using FlowJo software (TreeStar). Cells were gated for single and live cells. Cells were then gated on YFP +, which were also CD45+ indicating they are immune infiltrating cells. Immune cell populations were analysed by their expression of CD11b and Ly6C. YFP+ CD11b+ immune cells were MHC-II−, CD11c−, F4/80−, Ly6G−, CD19−, CD3− and NK1.1−. See Supplementary Fig. 12 for gating strategies.

### IFNβ RT-PCR on sorted cell populations

To analyse IFNβ gene expression in different cell subsets, AB1 tumours were treated with 50 ug of Poly(I:C) (HMW, Invivogen) i.t. ($n = 5$) or untreated ($n = 5$), 5 days after inoculation, then harvested 24 h later. Tumours were immediately submerged in cold PBS, cut into 1–2 mm pieces with a scalpel blade and dissociated using the GentleMACS system (Miltenyi). Fc block (anti-CD16/CD32, BD) was used for 10 minutes on ice. Cells were stained with UV Zombie live/dead (Biolegend) for 30 minutes at RT, to discriminate live cells. Cells were stained using antibodies for surface markers for 30 minutes at 4 °C. Cells were then sorted into RNAlater (Invitrogen) for the following populations: non-immune cells (CD45−); Ly6Chi monocytes (CD45+ CD11b+ Ly6Chi CD3− CD335−), Ly6Clo/− monocytes (CD45+ CD11b+ Ly6Clo/− CD3− CD335−), and the remaining immune cell (CD45+ CD11b−). RNA was extracted using the RNAqueous-Micro Kit (cat# AM1931, Life Technologies). The resulting purified RNA was reverse transcribed using a High-Capacity cDNA Reverse Transcription Kit (cat# 4368814, Applied Biosystems). Next, we performed RT-PCR using TaqMan Fast Advanced Master Mix (Applied Biosystems) and TaqMan Assay mouse IFNβ1 (Mm00439552_s1, ThermoFisher) or mouse GAPDH (Mm99999915_g1, ThermoFisher) in triplicate for each sample in a MicroAmp optical plate (Applied Biosystems) using QuantStudio 7 Flex Real-Time PCR System (Applied Biosystems). *Ifnb1* expression was calculated as dCT of the housekeeping gene GAPDH. ΔCt = Ct (*Ifnb1*) − Ct (GAPDH).

### Power calculation

The sample size calculation for in vivo mouse experiments was based on prior experiments in which we found that the median survival time on the control treatment (ICB alone) was 35 days[15,16]. Using a proportional hazards model we determined that, if the true hazard ratio (relative risk) of control subjects relative to experimental subjects is 5, we would need to study 10 experimental subjects and 10 control subjects to be able to reject the null hypothesis that the experimental and control survival curves are equal with probability (power) 0.8. The type I error probability associated with this test of this null hypothesis is 0.05.

### Statistics

Differences in population frequencies in responders and non-responders using flow cytometry were assessed using Mann-Whitney U testing on means. Prism software (GraphPad) was used to analyse tumour growth and for statistical significance of differences between groups by applying a Mann−Whitney U test. P-values were adjusted for multiple comparisons using the Benjamini-Hochberg (B-H) method; those <0.05 were considered significant. The Kaplan-Meier method was used for survival analysis, and p-values were calculated using the log-rank test (Mantel−Cox). For comparison of tumour size when complete response was not achieved, growth curves were analysed using the TumGrowth package[91]. For comparison of deconvolution estimations, we used two-way ANOVA with Tukey's multiple comparisons test.

### Reporting summary

Further information on research design is available in the Nature Research Reporting Summary linked to this article.

## Data availability

The RNA-Seq generated in this study have been deposited in the Gene Expression Omnibus database under accession code GSE153941 for the bulk RNAseq data, and GEO: GSE153942 for the single cell RNAseq data. The human breast cancer single-cell data (https://doi.org/10.1038/s41591-021-01323-8) used in this study is publicly available to download as read count data per individual patient at http://biokey.lambrechtslab.org. Source data are provided with this paper.

## Code availability

Code for the analysis in the main manuscript and the supplementary data is available through GitHub (https://github.com/wlchin/IFNsignalling, https://doi.org/10.5281/zenodo.6635312).

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

## Acknowledgements

We thank Michael Small, Ayham Zaitouny, Emma de Jong and Leo Portes dos Santos for discussions and Ian Dick for statistical support. This work was funded by NHMRC grants 1103980 and 1107043. W.L.C. was supported by a scholarship from the ADSA and a WA DoH Cancer and Palliative Care Network Fellowship. R.M.Z. was supported by a Forrest Prospect Fellowship. T.L. was supported by a Feilman Foundation fellowship. W.J.L. was supported by a Simon Lee fellowship, an NHMRC fellowship and a Cancer Council WA fellowship. A.R.R.F. is supported by an NHMRC fellowship, APP1154524. Part of this research was made possible by the Australian Cancer Research Foundation Centre for Advanced Cancer Genomics, a collaborative cancer research grant provided by the Cancer Research Trust and an enabling grant from Cancer Council WA.

## Author contributions

R.M.Z, W.L.C, T.L and W.J.L designed research; R.M.Z., V.S.F., B.W., T.H.C., C.F., C.M.T. and B.G. performed experimental work; W.L.C, R.M.Z, A.B., A.R.R.F., M.J.M., A.K.N., R.A.L., T.L and W.J.L analysed the data; L.B. contributed new reagents; R.M.Z, W.L.C., T.L. and W.J.L. wrote the manuscript. T.L and W.J.L supervised and managed the overall study.

## Competing interests

Patent application pertaining to aspects of this work: PCT/AU2021/050764. W.J.L.: consultancy Douglas Pharmaceuticals, MSD, research funding Douglas Pharmaceuticals, AstraZeneca, ENA therapeutics. A.K.N. advisory boards Boehringer Ingelheim, Bayer, Roche, BMS; research funding from AstraZeneca. M.M.: Advisory boards MSD, BMS, Roche, AstraZeneca. The remaining authors declare no competing interests.
