## [Peer Review File · Nature Communications]

Temporally restricted activation of IFN β signaling underlies response to immune checkpoint therapy in miceREVIEWER COMMENTS

Reviewer #1 (Remarks to the Author):

In this manuscript, Rachael M. Zemek et al. perform a longitudinal RNA-seq analysis of tumor samples obtained from two bilateral mouse tumor models treated with anti-PD-L1 and anti-CTLA-4. The bilateral models allow them to analyze one of the tumors while following tumor growth in the contralateral tumor to classify the mouse as responder or non-responder. The authors identify a cluster of interferon response genes that show a different behavior in responders versus non-responders. The genes in this cluster are highly expressed before treatment and rapidly decrease after therapy in the AB1 tumor model. In the case of the RENCA tumor model, the gene expression of these genes peaks two days after treatment. This on-off kinetics is experimentally mimicked by administration of poly(I:C) followed by the combination of anti-PD-L1 and anti-CTLA-4 and then anti-IFNAR or anti-IFN- β . Using single-cell RNAseq, the authors identified proinflammatory monocytes as the subset of immune cells that exhibit this on-off IFN- β expression. Finally, the on-off gene signature is detected in myeloid cells from breast cancer samples before anti-PD-1 treatment in those patients that expand tumor-infiltrating T cells.

The combination of anti-PD-1 and anti-CTLA-4 is the first combination of two immune-oncology agents that has been approved for the treatment of different malignancies. This manuscript identifies a novel immune mechanism that can be clinically actionable and potentially may enhance the percentage of patients who benefit from immunotherapy. The use of antibodies against interferon-beta to enhance the antitumor activity of immune checkpoint is an original and novel finding, in line with other reports that used other anti-inflammatory drugs. The manuscript highlights the relevance of kinetics in the immune-oncology interventions that is often neglected due to the complexities of the clinical implementation of this approach, but that may be extremely relevant for further advances in immuno-oncology. The methodology is elegant and well described and allowed the authors to identify the subtype of type I interferon and the cellular source responsible for the on-off kinetics.

Major comments:

- 1) In page 4, the authors highlight the differences between responders and non-responder in cluster 4 but tend to minimize the differences observed in cluster 2. Since the authors use a “model containing time-point and response as covariates using a likelihood ratio test”, the results of this statistical analysis should be reported, and, therefore, an objective comparison could be provided. While cluster 4 may be of great interest due to the early changes detected, the huge induction of cluster 2 genes may reflect the immune mechanisms that ultimately are able to eradicate the tumors.
- 2) Regarding cluster 4, important differences are observed between the AB1 and RENCA. In the AB1 tumor model, the genes related to interferon are induced 1 hour before therapy and quickly decrease, while in the RENCA model, there are no baseline differences between responder and non-responders. These differences have important clinical implications and must be recognized in the text. Based on the AB1 tumor, the endogenous activation of the type I interferon may be critical and, therefore, detectable in biopsies obtained before therapy. However, the kinetics observed in the RENCA tumors seems to signal an early modulation of the tumor microenvironment by the immune checkpoint combination. So, the question that arises is whether the kinetics in both models are independent or dependent on the immune checkpoints? The kinetics of gene expression of several genes of cluster 4 in untreated tumors may be highly informative.
- 3) One of the most relevant findings of this manuscript is the on/fast-off signature that is used in Figure 5J, but a detailed description of the definition of the gene signature is not provided. What genes belong to this signature? How is the kinetics information implemented in this signature? Or, in other words, what is the difference between a type I interferon signature and the proposed on/fast-off signature?
- 4) In the last section of the Results, published single cell RNAseq data from breast cancer patients treated with anti-PD-1. Data from pre- and post-treatment biopsies are used. The authors claim that the on/fast-off IFN kinetics in monocytic cells correlate with response. It is important to clarify that the term response does not mean clinical response since clinical data is not reported in the original manuscript. The authors should refer to this response as immune response or something similar. Certainly, it would be desirable that a correlation with clinical response is shown.

5) The original document that reported the data from breast cancer patients identified an upregulation of several interferon-related genes in macrophages (see figure 5I or reference 24). What is specific about the new on/fast IFN signature? Is this signature superior to the genes identified in reference 24? And is this on/fast IFN signature also present in macrophages in addition to CCR2+ monocytes?

6) The on/off kinetics identified in the longitudinal RNAseq analysis is fast. The main differences are observed within two days. However, the experimental protocol to mimic this on/off kinetics is based on daily doses of poly(I:C) for three days, followed by immune checkpoints three days after the final dose of poly(I:C) and the anti-IFNAR monoclonal antibody is administered three days after the first day of immune checkpoint administration. Therefore, nine days separate the initiation of the interferon induction from the administration of the first dose of the antibody to block the type I interferon receptor. What is the reason for the discrepancy in the time scale? Based on the RNAseq data, an earlier administration of the antibody to block IFNAR should be better? How was the best protocol determined?

Reviewer #2 (Remarks to the Author):

General comments :

Zemek et al described a dynamic modulation of type I IFN signature related to ICB responses by using a bilateral tumor syngeneic mouse model developed by the same laboratory (Nat Protocol, 2020). By using two different mouse tumor models, the authors found that the dynamics of monocyte-derived IFNbeta in the TME is a critical factor determining the tumor response to ICB treatment. That saying, a fast ON/OFF IFNbeta and/or related signaling is prerequisite to a good prognostic for ICB, while sustained (thus chronic) stimulation of the TME by IFNbeta mitigates the benefit of ICB, thus leading to resistance and relapse.

The current study is in general well designed, the extended application of their bilateral tumor model in the ICB resistance is very interesting and may be useful for the immuno-oncology research domain. It is also highly appreciated that the authors nicely showed large amount of in vivo experiments with IFN blocking strategies. However, the current findings are not really novel. The chronic exposure to IFN in the TME is already known to be an important factor for the development of resistance to ICB, for example those works published in Benci et al, Cell, 2016 ; Jacquelot et al, Cell Res, 2019 etc. Apart from this, I believe that the most interesting contribution of the current work is using a bilateral tumor model to examine the early dynamic changes of TME with a power to predict the later response during ICB treatment. And some other critical viewpoints need to be further clarified :

Major points :

- 1) No matter what is the response, the authors have observed increased infiltration of CD8+ T cells in the two used tumor models, although with higher fractions of CD8+ T cells in the responders. While the population of monocyte that the authors are focusing on in the current work is actually also increased in all the conditions with even higher magnitude compared the increase of CD8+ T cells. It is not clear whether the monocyte derived IFNbeta is determinant for the cell state of infiltrated CD8+ T cells. Whether the infiltrated CD8+ T cells are CD8+PD1+ in the responders and CD8+PD1+TIM3+/LAG3+ in the non-responders ? If the infiltration of CD8+ T cells are in different state between responders and non-responders, whether it is a consequence of the chronic exposure of IFNbeta in the non-responders ?
- 2) It is quite interesting to dissect the early changes of the TME during ICB treatment by using this bilateral tumor model. However, at the late stage of the tumor growth (ie. Around 15 -20 days), do the authors observe similar differential fractions of monocyte between responders and non-responders ? What is the concentration of IFNbeta in the interstitial liquid of corresponding tumors ? and what is the status of the key TF IRF1 in the responder/non-responder tumors ? Without the analysis of the late stage of the tumors, it is difficult to conclude with early dynamic analysis to predict the later-on response. It will be very nice to show that the imprints of chronic exposure of early dynamics of IFNbeta on the differential TME in the later stage.
- 3) AB1 and Renca seemly show a totally different basal innate immunity, with AB1 showing a very

high basal IFN activity before treatment. This may be correlated with the much higher fraction of macrophage at the basal level in AB1 tumors. It is known that macrophage derived IFNbeta/IFNgamma is important for the resistance to ICB treatment through inducing Nos2. Do the authors have any data showing that macrophages are much less implicated in the current models ? In addition to this, the same authors have published in Science translational medicine 2019 that NK cells are implicated to the response to ICB, in that study, they showed a totally different proportion of NK cells and CD8+T cells. Could they further discuss the discrepancies from the two studies, and whether the IFNbeta exposure is related to NK function that may also be predictable for later stage response to ICB in the current study ?

4) In the figure 4A, could the authors show clearly what are the related regulators in different clusters ? It seems that there are quite different dynamics of different regulators. Are the fast ISG regulators in the second cluster ? and the third big cluster shows also differential dynamics between responders and non-responders, with concurrent higher level in responders from day 4-6 in both models. Are they powerful to predict the later-on response ?

5) In the figure 5C-D, it is confusing that IFNbeta-stimulated T cell signature is much higher in non-responders. Related to their figure 1, can we think these T cell signatures are related to an exhausted T cell state ?

6) The patient data is not quite enough to consolidate their in vivo observations. Although the authors nicely demonstrated that they can find a similar circulating-derived CCR2+ monocyte is present in patient samples. It is not clear what are the dynamic changes of IFNbeta in these human samples. In addition, why this population is higher in expanded patients compared non-expanders ? Does this signify that there are more production of IFNbeta in the expanders ?

Minor points :

1) The authors should show statistical evaluation in the whole study. For example, figure 1D, figure 5B, 5J, 5K, and some supplemental figures.

2) What is the distribution of each cluster of the single cell RNAseq study between responders and non-responders in figure 5A ?

3) By using the IFNbeta-YFP mice, can the authors show that there are indeed different proportions of CD11b+Ly6C+IFNbeta+ monocytes between responders and non-responders ?

4) Some typo should be corrected, for example Page 7 line 220 « responders using ».

Reviewer #3 (Remarks to the Author):

This manuscript from Zemek et al. describes a critical role for time dependent signaling from IFNbeta in murine models of checkpoint blockade immunotherapy. The authors use computational deconvolution of bulk RNAseq data validated by single cell RNAseq and therapeutic interventions to show that IFNbeta produced by myeloid cells early in an immune response to cancer enhances the efficacy of checkpoint blockade, but that these responses are significantly augmented by rapid clearance of the IFNbeta signal. A particularly strength of their data is the multiple time points captured, and their ability to distinguish responders from non responders by stochastic factors rather than through differences in treatment approach – this is made possible by an innovative two tumor model, where the analyzed tumor is removed for analysis and the remaining tumor is used to determine response. This model has been previously published by the authors, but is nevertheless of interest. Overall, the data are convincing, and my concerns are more about how well the authors have established the generalizability of their findings.

Substantive Issues

1. Although the authors have used more than one cell line, they did not look in any of the very common tumor cell lines used by other groups studying immunotherapy responses (B16F10, MC38, CT26, for example) and they did not provide any compelling reason why these models were not studied. In particular, they do not have any data from C57BL/6 mice. While such experiments are not necessary, the absence of these models does reduce the authors ability to make broad generalizations about the mechanism they have uncovered. I strongly suggest that the introduction and conclusion be narrowed to indicate that this is a potentially important mechanism that has been

validated to some extent in the models examined but that awaits further testing in a broader range of systems.

2. Similarly, the authors used a narrow set of immunotherapies, and chose to treat all of their mice with anti-CTLA-4/anti-PD-L1, a combination that is not used clinically. In addition, CTLA-4 targeting therapies in mice appear to deplete Tregs, a function that does not clearly underly efficacy in humans. As with the comment above, this limitation just needs to be noted because it may limit the translatability of these findings.

3. Overall, the data on RENCA cells are less convincing than with the other cell lines. This fact makes point #1 that much more important.

4. The human data are confusing and, at least for me, require more explanation. How precisely were the human “on/fast-off” signatures defined at a single time point. Were the authors looking at homologous genes that had been identified through the murine analysis, and presuming that they turned on a off, or did they use a pseudotime analysis? I am not sure that a pseudotime analysis could be considered valid here in any case, because the sample was taken prior to immunotherapy, and the authors demonstrated in mice that IFN β signaling should be on in tumors at the time of checkpoint blockade therapy, and that the process of turning off this pathway has to happen later in the treatment course to be beneficial. Simply indicating that IFN signaling is found in future responders to immunotherapy is not surprising, and is not specific enough to the authors mechanisms for them to consider this human sample validation.

5. Breast tumors also respond very poorly to immunotherapy in general, and so any findings in those tumors may be of limited value for understanding the immune response in tumors that respond more readily such as melanoma. I understand that the authors had these data available, but this limitation coupled with #4 should make them significantly temper their statements about validation in human samples, and I am not even convinced at this point that these data belong in this manuscript.

I have a few small suggestions as well

6. Figure 3A really doesn't provide any information without having any of the genes labeled, which obviously won't be legible in a main figure. I suggest that this figure be moved to the supplement and that gene labels be added on a larger image.

7. Figure 3B and 3C should be enlarged – at the current size, I really cannot read what the authors are showing

8. On the survival curves, it would be nice to have the timing that treatments were given marked on the x-axis

9. Figure 4C and 4D may benefit from another method to validate the findings, as is the validity of the results are difficult to gauge – perhaps isolating the T cells or performing single cell sequencing at specific time points so that the authors are sure that the IFN response signal is indeed T cell derived

10. For figure 5I, it would be helpful to see the same plot for the YFP negative cells in order to assess the validity of the determined gates. A similar analysis should be coupled with the supplemental figure that shows the overall gating strategy.

POINT-BY-POINT REPLY TO REVIEWER COMMENTS

We thank the reviewers for their constructive comments and feedback on our manuscript. Based on their comments, we have performed additional experiments and analyses, providing further experimental evidence of the crucial role of rapid on/off IFN β activation in the response to immune checkpoint therapy, which has significantly improved the manuscript.

Reviewer #1 (Remarks to the Author):

In this manuscript, Rachael M. Zemek et al. perform a longitudinal RNA-seq analysis of tumor samples obtained from two bilateral mouse tumor models treated with anti-PD-L1 and anti-CTLA-4. The bilateral models allow them to analyze one of the tumors while following tumor growth in the contralateral tumor to classify the mouse as responder or non-responder. The authors identify a cluster of interferon response genes that show a different behavior in responders versus non-responders. The genes in this cluster are highly expressed before treatment and rapidly decrease after therapy in the AB1 tumor model. In the case of the the RENCA tumor model, the gene expression of these genes peaks two days after treatment. This on-off kinetics is experimentally mimicked by administration of poly(I:C) followed by the combination of anti-PD-L1 and anti-CTLA-4 and then anti-IFNAR or anti-IFN- β . Using single-cell RNAseq, the authors identified proinflammatory monocytes as the subset of immune cells that exhibit this on-off IFN- β expression. Finally, the on-off gene signature is detected in myeloid cells from breast cancer samples before anti-PD-1 treatment in those patients that expand tumor-infiltrating T cells. The combination of anti-PD-1 and anti-CTLA-4 is the first combination of two immune-oncology agents that has been approved for the treatment of different malignancies. This manuscript identifies a novel immune mechanism that can be clinically actionable and potentially may enhance the percentage of patients who benefit from immunotherapy. The use of antibodies against interferon-beta to enhance the antitumor activity of immune checkpoint is an original and novel finding, in line with other reports that used other anti-inflammatory drugs. The manuscript highlights the relevance of kinetics in the immune-oncology interventions that is often neglected due to the complexities of the clinical implementation of this approach, but that may be extremely relevant for further advances in immuno-oncology. The methodology is elegant and well described and allowed the authors to identify the subtype of type I interferon and the cellular source responsible for the on-off kinetics.

Major comments:

1) In page 4, the authors highlight the differences between responders and non-responder in cluster 4 but tend to minimize the differences observed in cluster 2. Since the authors use a “model containing time-point and response as covariates using a likelihood ratio test”, the results of this statistical analysis should be reported, and, therefore, an objective comparison could be provided. While cluster 4 may be of great interest due to the early changes detected, the huge induction of cluster 2 genes may reflect the immune mechanisms that ultimately are able to eradicate the tumors.

The reviewer is correct in pointing out there are differences in magnitude observed between responders and non-responders in cluster 2. Both cluster 1 and cluster 2 show an increase in immune related genes over time. However, this trend is the same in responders and non-responders. We now included an additional supplementary figure (S2) showing that cluster 2 is in fact the most similar between the two phenotypes. Only cluster 4 displayed different dynamics when comparing RS and NR across both models and was therefore of most interest. Furthermore, these differential dynamics could only be identified using time course data in a controlled setting, i.e. not from patient samples, demonstrating the model's unique effectiveness in identifying differences in gene dynamics.

The DE genes from of the likelihood ratio test are given in file Data S1.

2) Regarding cluster 4, important differences are observed between the AB1 and RENCA. In the AB1 tumor model, the genes related to interferon are induced 1 hour before therapy and quickly decrease, while in the RENCA model, there are no baseline differences between responder and non-responders. These differences have important clinical implications and must be recognized in the text. Based on the AB1 tumor, the endogenous activation of the type I interferon may be critical and, therefore, detectable in biopsies obtained before therapy. However, the kinetics observed in the RENCA tumors seems to signal an early modulation of the tumor microenvironment by the immune checkpoint combination. So, the question that arises is whether the kinetics in both models are independent or dependent on the immune checkpoints? The kinetics of gene expression of several genes of cluster 4 in untreated tumors may be highly informative.

Previously, we compared gene expression data from responders and non-responders to CTLA4 treatment in AB1 on day 6 after treatment, which included untreated controls as well (Lesterhuis et al. Sci Rep 2015). We found that the non-responders and untreated mice clustered completely together. In addition, in ongoing unrelated studies, we have characterised the untreated tumour microenvironment of two other tumour models in BALB/c mice *in time*, WEHI-164 and CT-26, which showed no effects on the expression of IFN-related genes in time. Together, this demonstrates that the observed effects relating to cluster 4 genes are indeed dependent on checkpoint therapy and not a feature of a growing s.c. tumour.

However, the reviewer rightly points out that there are some differences between Renca and AB1, particularly prior to treatment, as we also discussed in the paper. This is not unlike the clinical situation where some correlates with response are more pronounced in some cancers than others, as for example has been reported for a pre-treatment IFN γ gene expression signature, showing a significant correlation with best overall response in melanoma or head and neck cancer, but not in gastric cancer (Ayers et al, JCI 2017). We have now expanded our discussion of these differences in the paper on page 4.

3) One of the most relevant findings of this manuscript is the on/fast-off signature that is used in Figure 5J, but a detailed description of the definition of the gene signature is not provided. What genes belong to this signature? How is the kinetics information implemented in this signature? Or, in other words, what is the difference between a type I interferon signature and the proposed on/fast-off signature?

We have now provided our fast-off gene list in a supplementary data file (Data S5). In addition, we show how our signature is constructed from the interferon response hallmarks gene sets in figure S3F.

4) In the last section of the Results, published single cell RNAseq data from breast cancer patients treated with anti-PD-1. Data from pre- and post-treatment biopsies are used. The authors claim that the on/fast-off IFN kinetics in monocytic cells correlate with response. It is important to clarify that the term response does not mean clinical response since clinical data is not reported in the original manuscript. The authors should refer to this response as immune response or something similar. Certainly, it would be desirable that a correlation with clinical response is shown.

We resolved any ambiguity by now referring to these groups in the paper as “T cell expanders” or “T cell non-expanders”. We used the only available clinical dataset that has paired samples before and after anti-PD1. Although these patient samples do not provide the temporal granularity of four time points or definitive clinical response data, like our murine study does (which was designed as such for this reason), they do highlight a role for monocytes in the response to ICB. We have now further emphasized these limitations on page 11

5) The original document that reported the data from breast cancer patients identified an upregulation of several interferon-related genes in macrophages (see figure 5I or reference 24). What is specific about the new on/fast IFN signature? Is this signature superior to the genes identified in reference 24? And is this on/fast IFN signature also present in macrophages in addition to CCR2+ monocytes?

The paper by Bassez et al uses previously described macrophage phenotypes from Qian et al (2020, Nature), which classify CCR2⁺ myeloid cell as early stage/intermediate monocyte-macrophages. Blood-derived monocytes (which express CCR2⁺) that encounter the tumour microenvironment alter their phenotype (becoming the pro-inflammatory population described) and begin the transition to a macrophage phenotype. The line between monocyte and macrophage is subjective; in our data they express the M1 marker CXCL9 (by single cell analysis) and therefore could be considered a macrophage, however we classify them as monocytes due to the lack of F4/80 protein expression by flow cytometry, and a more immature phenotype suggested by RNA trajectory analysis.

It is difficult to directly compare signatures, as our fast-off signature is not predictive at all time points. Rather, it is a dynamic on/off change, and we do not yet know the dynamics for patients or different tumour types.

Furthermore, the signature in reference 24 (now reference 25) is the result of identifying differentially expressed genes in the authors’ own dataset, and therefore will be the most predictive of their own dataset. Additionally, there are multiple metrics for what constitutes a superior gene set, depending on both the dataset and the biological question at hand. Considering these caveats, we can demonstrate that our fast-off signature (1) has a higher normalised enrichment score when comparing CCR2⁺ myeloid cells between expanders and non-expanders and (2) ranks statistically higher than the ref. 24 gene set in pathway overdispersion analysis. See Data S7 and Figure S10e:

We interpret these metrics to (1) support the generalisability of our signature to human data and to indicate that (2) our signature is more relevant to the ICB response than a generic interferon signature revealed through exploratory analysis of a human single-cell dataset of ICB-treated tumours. We added this to the text on page 9.

As seen in fig 5C (mouse data) and S10 (human data), the signature is not restricted to one population. It is also expressed in other myeloid populations, but it is most enriched in the monocyte populations when comparing responders with non-responders. Using flow cytometry in our B6.129-Ifnb1^{tm1Lky}/J mice, comparing responders and non-responders, we now unequivocally demonstrate that (F4/80 negative) inflammatory monocytes in the tumour expressed IFN β (YFP⁺), enriched in responders. We added this data to Figure 5.

6) The on/off kinetics identified in the longitudinal RNAseq analysis is fast. The main differences are observed within two days. However, the experimental protocol to mimic this on/off kinetics is based on daily doses of poly(I:C) for three days, followed by immune checkpoints three days after the final dose of poly(I:C) and the anti-IFNAR monoclonal antibody is administered three days after the first day of immune checkpoint administration. Therefore, nine days separate the initiation of the interferon induction from the administration of the first dose of the antibody to block the type I interferon receptor. What is the reason for the discrepancy in the time scale? Based on the RNAseq data, an earlier administration of the antibody to block IFNAR should be better? How was the best protocol determined?

We chose the timing of the antibody treatments solely based on the kinetics as observed in the RNAseq data, which is ON before ICB and OFF 3 days after ICB. With regards to the ON signal, we previously found that dosing poly(I:C) for 3 days prior to ICB gave a priming effect in both AB1 and Renca, mimicking the strong IFN signal, also considering that poly(I:C) takes time to induce production of IFN proteins which also must exert their effects (Zemek et al. *Sci Transl Med* 2019).

The OFF signal starts 3 days after the first dose of ICB. We therefore selected this timepoint to initiate anti-IFN.

We found that when anti-IFNAR was initiated before ICB, the response was abrogated (Fig S4E, F), demonstrating that type I IFN is indeed required in the early phase of the therapeutic response.

.....

Reviewer #2 (Remarks to the Author):

General comments :

Zemek et al described a dynamic modulation of type I IFN signature related to ICB responses by using a bilateral tumor syngeneic mouse model developed by the same laboratory (Nat Protocol, 2020). By using two different mouse tumor models, the authors found that the dynamics of monocyte-derived IFN β in the TME is a critical factor determining the tumor response to ICB treatment. That saying, a fast ON/OFF IFN β and/or related signaling is prerequisite to a good prognostic for ICB, while sustained (thus chronic) stimulation of the TME by IFN β mitigates the benefit of ICB, thus leading to resistance and relapse.

The current study is in general well designed, the extended application of their bilateral tumor model in the ICB resistance is very interesting and may be useful for the immuno-oncology research domain. It is also highly appreciated that the authors nicely showed large amount of in vivo experiments with IFN blocking strategies. However, the current findings are not really novel. The chronic exposure to IFN in the TME is already known to be an important factor for the development of resistance to ICB, for example those works published in Benci et al, Cell, 2016 ; Jacquelot et al, Cell Res, 2019 etc. Apart from this, I believe that the most interesting contribution of the current work is using a bilateral tumor model to examine the early dynamic changes of TME with a power to predict the later response during ICB treatment. And some other critical viewpoints need to be further clarified :

We thank the reviewer for pointing out the data in these papers, which we had indeed referenced in the manuscript. There is an important distinction between those papers and our results. Benci et al (Cell, 2016) and Jacquelot et al (Cell Res, 2019) have shown that *secondary* ICB-resistant cancers, which are cancers that initially responded but then relapsed, display chronically active IFN signaling, specifically IFN γ . This is not the case in our model which uses *intrinsically responsive* tumors. In this treatment-naïve setting, which is clinically the most prevalent and relevant setting, we found that type I IFN only (not type II), and more specifically IFN β not α , plays a dual role in a time-dependent manner. In addition, our data do not indicate that chronic IFN β causes non-response, as such; amplitude and timing are both important. High IFN β activity, followed by rapid switching off are both required to respond, as we show experimentally. Together, these results provide a new insight into the type of IFN (type I not type II; β not α), the amplitude of IFN activity and the kinetics that underlie the response to ICB, compared to the papers in the literature. We clarified these aspects now on page 10/11 in the discussion.

Major points :

1) No matter what is the response, the authors have observed increased infiltration of CD8+ T cells in the two used tumor models, although with higher fractions of CD8+ T cells in the responders. While the population of monocyte that the authors are focusing on in the current work is actually also increased in all the conditions with even higher magnitude compared the increase of CD8+T cells. It is not clear whether the monocyte derived IFN β is determinant for the cell state of infiltrated CD8+ T cells. Whether the infiltrated CD8+ T cells are CD8+PD1+ in the responders and CD8+PD1+TIM3+/LAG3+ in the non-responders ? If the infiltration of CD8+ T cells are in different state between responders and non-responders, whether it is a consequence of the chronic exposure of IFN β in the non-responders ?

We did not find a significant difference in either CD8⁺ T cells or Monocytes between RS and NR at any timepoint (we added 'n.s.' to figure 1D). Both RS and NR increase CD8⁺ T cells compared to baseline, making it an unreliable biomarker.

Results for CD8⁺ T cells comparing RS to NR, or time-points to baseline (supp data 2):

ANOVA (Non-parametric, Kruskal-Wallis test with Benjamini Hochberg correction for multiple comparisons)		
CD8 T cells		
Group	comparison	q value
AB1 day 0	RS vs NR	0.2596
AB1 day 2	RS vs NR	0.1802
AB1 day 4	RS vs NR	0.1802
AB1 day 6	RS vs NR	0.2087
Renca day 0	RS vs NR	0.681
Renca day 2	RS vs NR	0.3824
Renca day 4	RS vs NR	0.2596
Renca day 6	RS vs NR	0.1802
AB1 RS	0 vs. 2	0.0121
AB1 RS	0 vs. 4	<0.0001
AB1 RS	0 vs. 6	<0.0001
AB1 NR	0 vs. 2	0.084
AB1 NR	0 vs. 4	0.0016
AB1 NR	0 vs. 6	<0.0001
Renca RS	0 vs. 2	0.1862
Renca RS	0 vs. 4	0.001
Renca RS	0 vs. 6	<0.0001
Renca NR	0 vs. 2	0.4961
Renca NR	0 vs. 4	0.023
Renca NR	0 vs. 6	0.0121

Although the proportion of T cells was not found to be significant, we agree with the reviewer that the phenotype of the cells could be a factor, particularly the expression of other checkpoints. We analysed the bulk RNAseq data to test whether responders had lower expression of checkpoints compared to non-responders (Fig. S2). We found that gene expression of checkpoint markers TIM3, LAG3, OX40, PD1 and PDL2 was higher in responders and increased over the course of therapy, suggesting that activation state rather than their numbers per se were more predictive of a response. Figure S2:

To further define how IFN β modulates T cell responses, we performed flow cytometry on tumours. To determine the role of chronic vs. on/off IFN β activity, we dosed tumours with either recombinant murine IFN β for 6 days (chronic) or for 3 days followed by an anti-IFN β antibody, and analysed tumours by flow cytometry to determine checkpoint expression on T cells (Fig. S7):

We found that on/off IFN β signalling resulted in enhanced recruitment of CD4 $^+$ and CD8 $^+$ T cells, compared to untreated or chronic IFN β . These T cells were also highly proliferative as evidenced by enhanced Ki67 expression, compared to controls. While having little effect on the expression of checkpoints TIM3 and LAG3, on/off IFN β activity did upregulate PD1 expression. This shows that the dynamics of IFN β signalling early-on during ICB affect T cell recruitment and PD1 expression in the tumour microenvironment. These data have been added as figure S7 and discussed in the text on page 8.

We could also see enhanced Ki67 expression of CD8 $^+$ T cells when on/off IFN β modulation was combined with ICB. However, there were no significant changes compared to ICB alone.

Figure: Flow cytometry of tumours showing Ki67+ CD8 T cells as % of CD45 after IFN β /ICB modulation

2) It is quite interesting to dissect the early changes of the TME during ICB treatment by using this bilateral tumor model. However, at the late stage of the tumor growth (ie. Around 15 -20 days), do the authors observe similar differential fractions of monocyte between responders and non-responders ? What is the concentration of IFNbeta in the interstitial liquid of corresponding tumors ? and what is the status of the key TF IRF1 in the responder/non-responder tumors ? Without the analysis of the late stage of the tumors, it is difficult to conclude with early dynamic analysis to predict the later-on response. It will be very nice to show that the imprints of chronic exposure of early dynamics of IFNbeta on the differential TME in the later stage.

In the later stages of these models, responding tumours are actively regressing, and by day 20 tumours are either too small to analyse or have completely regressed (see figure S1). Day 6 post-ICB is the latest time point we can analyse tumours before there is a noticeable size difference between responders and non-responders.

To address the reviewer's comment, we determined the number of IFN β expressing cells in responders and non-responders using the bilateral tumour model in our B6.129-Ifnb1^{tm1Lky}/J reporter mice, and analysed tumours 2 days after ICB by flow cytometry (total of 4 responders and 2 non-responders). Indeed, responders had higher numbers of IFN β -producing cells, in particular in the CD11b $^+$ CCR2 $^+$ Ly6C $^+$ monocyte fraction, confirming the single cell RNAseq data on the protein level. We added this data to figure 5:

Regarding the imprinting of the TME by early differences in IFN β activity, we refer to the data shown in reply to the previous question, demonstrating enhanced infiltration of highly proliferative CD4⁺ and CD8⁺ T cells following on/off IFN β signalling.

The key TF's Irf1, Irf7 and Stat1 are detected in the RNAseq data, and follow the cluster 4 kinetics (Data S3). Furthermore, Irf1 and Stat1 are upregulated in responder monocytes and macrophages in the single cell data. In figure 5F, cluster 1 monocytes have the strongest expression of these TFs. In figure S8C, velocity analysis shows Irf1 expression is being switched off rapidly in responder monocytes.

3)AB1 and Renca seemly show a totally different basal innate immunity, with AB1 showing a very high basal IFN activity before treatment. This may be correlated with the much higher fraction of macrophage at the basal level in AB1 tumors. It is known that macrophage derived IFNbeta/IFNgamma is important for the resistance to ICB treatment through inducing Nos2. Do the authors have any data showing that macrophages are much less implicated in the current models ? In addition to this, the same authors have published in Science translational medicine 2019 that NK cells are implicated to the response to ICB, in that study, they showed a totally different propotion of NK cells and CD8+T cells. Could they further discuss the discrepancies from the two studies, and whether the IFNbeta exposure is related to NK function that may also predictable for later stage response to ICB in the current study ?

Yes, AB1 and Renca are very different in terms of their cellular infiltrate, yet both are responsive to ICB. It is interesting that their basal immunity also differs substantially, particularly that the macrophage dense tumour has the highest IFN signature before therapy. The on/off IFN β dynamics correlates with an M1-like gene signature, however there is little difference in the M2 signature:

Nos2 (iNOS) gene expression increases after ICB in *responders* in the Renca model, and therefore Nos2 is unlikely to be causing resistance (see figure S2i):

Both studies have the same result: NK cells are significantly higher in responders prior to treatment, and CD8⁺ T cells are not significantly different. Note that in the 2019 paper, CIBERSORT cell populations are shown as a “relative” proportion (i.e. all immune cells add up to 100%). In this paper we have used the most up to date version of CIBERSORT with updated reference matrix, and have presented the data as “absolute” (i.e. an estimate of the fraction of immune cells of the total sample). Figure S1d:

4) In the figure 4A, could the authors show clearly what is the related regulators in different clusters? It seems that there are quite different dynamics of different regulators. Are the fast ISG regulators in the second cluster? and the third big cluster shows also differential dynamics between responders and non-responders, with concurrent higher level in responders from day 4-6 in both models. Are they powerful to predict the later-on response?

Figure 3A gives an overview of the GENIE3 analysis and shows that there are different dynamics for regulators associated with the response to ICB. We have added the gene lists for each of the modules, now labelled on the figure (data S4). The third large module with a higher score on days 4-6 in responders is similar to clusters 1 and 2 in figure 2, containing genes related to IFN γ and the adaptive immune response. Although, as expected, this module was more activated in responders, it followed the same trend in non-responders, thus providing relatively low predictive value.

5) In the figure 5C-D, it is confusing that IFN β -stimulated T cell signature is much higher in non-responders. Related to their figure 1, can we think these T cell signatures are related to an exhausted T cell state?

In figure 4C-D we used a signature derived from IFN β stimulated T cells, which is the only comprehensive dataset of IFN β stimulated cells currently available. This dataset was then applied to our bulk RNAseq data, which is a mix of cells, and therefore is not T-cell specific as other immune cell types likely upregulate similar genes in response to IFN β . The figure demonstrates that the on/off signature is partially attributed to IFN β stimulation, as opposed to IFN γ . We have added a clarification to page 7.

6) The patient data is not quite enough to consolidate their in vivo observations. Although the authors nicely demonstrated that they can find a similar circulating-derived CCR2 $^+$ monocyte is present in patient samples. It is not clear what is the dynamic changes of IFN β in these human samples. In addition, why this population is higher in expanded patients compared non-expanders? Does this signify that there are more production of IFN β in the expanders?

The human dataset was used to validate the murine findings of the IFN β on/off signature derived from tumour-infiltrating monocytes as a driver of response, and we conclude these CCR2 $^+$ monocytes are "the predominant source of type I IFN activity". Expanders do show an increase in CCR2 $^+$ monocytes after treatment, which is reported in the original paper. We demonstrate that these cells are responsible for the fast-off IFN signature, and that expanders have higher expression of this signature pre-treatment within this population (regardless of

number of cells). However, as we do not have access to longitudinal tumour samples from patients undergoing ICB with both responders and non-responders, we cannot yet map these dynamics in patients. We emphasized this in the discussion on page 11.

As the data includes pre- and on-treatment ICB samples, we have expanded Figure S10d:

Minor points :

1)The authors should show statistical evaluation in the whole study. For example, figure 1D, figure 5B, 5J , 5K, and some supplemental figures.

Thank you for your comment. We have now added statistics to the figures 1D and S1D where appropriate. We have incorporated additional statistical analyses to highlight the biological relevance of the signatures to the ICB response in both datasets. Briefly, the statistical significance of “global” pathway enrichment (indicating that the pathway is differentially activated in the dataset) in both the murine and human data is now described in Data S7. The magnitude of cell-specific differential pathway enrichment (comparing responders vs. non responders across cell types) is shown in Data S7. We have updated the methods section to reflect these additional analyses on page 28.

2)What is distribution of each clusters of the single cell RNAseq study between responders and non-responders in figure 5A ?

The distribution of clusters is provided in Data S6. No statistically significant differences ($fdr < 0.05$, automatic cell reference) in monocyte proportions were detected between responders and non-responders at day 0 (determined using the scCODA package).

3)By using the IFNbeta-YFP mice, can the authors show that there are indeed different proportions of CD11b+Ly6C+IFN β + monocytes between responders and non-responders ?

See above reply to question 2 - We used the bilateral tumour model in B6.129-Irfn1^{tm1Lky}/J reporter mice, and analysed tumours 2 days after ICB by flow cytometry (total of 4 responders and 2 non-responders), showing indeed, responders had more IFN β ⁺ cells, and these cells were CD11b⁺CCR2⁺Ly6C⁺ monocytes.

4)Some typo should be corrected, for example Page7 line 220 « respondersusing ».

Thank you. We have corrected this.

.....

Reviewer #3 (Remarks to the Author):

This manuscript from Zemek et al. describes a critical role for time dependent signaling from IFNbeta in murine models of checkpoint blockade immunotherapy. The authors use computational deconvolution of bulk RNAseq data validated by single cell RNAseq and therapeutic interventions to show that IFNbeta produced by myeloid cells early in an immune response to cancer enhances the efficacy of checkpoint blockade, but that these responses are significantly augmented by rapid clearance of the IFNbeta signal. A particularly strength of their data is the multiple time points captured, and their ability to distinguish responders from non responders by stochastic factors rather than through differences in treatment approach – this is made possible by an innovative two tumor model, where the analyzed tumor is removed for analysis and the remaining tumor is used to determine response. This model has been previously published by the authors, but is nevertheless of interest. Overall, the data are convincing, and my concerns are more about how well the authors have established the generalizability of their findings.

Substantive Issues

1. Although the authors have used more than one cell line, they did not look in any of the very common tumor cells lines used by other groups studying immunotherapy responses (B16F10, MC38, CT26, for example) and they did not provide any compelling reason why these models were not studied. In particular, they do not have any data from C57BL/6 mice. While such experiments are not necessary, the absence of these models does reduce the authors ability to make broad generalizations about the mechanism they have uncovered. I strongly suggest that the introduction and conclusion be narrowed to indicate that this is a potentially important mechanism that has been validated to some extent in the models examined but that awaits further testing in a broader range of systems.

As published in Nature Protocols (Zemek et al 2020), when developing the bilateral model, there were several requirements: 1) the cell lines had to display dichotomous responses to ICB, which 2) were symmetrical, and 3) were of sufficient size at start of treatment to allow for enough tumour material to be obtained for RNAseq

analysis. Although we also tested cell lines B16, CT26, MC38, Line-1 and AE17, in addition to AB1 and Renca, they did not meet all these criteria.

Notwithstanding, Renca has been well-characterised and frequently used as a model for the response to ICB, for example by GlaxoSmithKline (Yu 2018, PloS One), Pfizer (Zhong 2020, BMC Genomics), MedImmune/AstraZeneca (Mosely 2017, Can Imm Res) and Merck (Georgiev, Mol Cancer Ther 2022).

In addition, we tested our on/off IFN β schedule in C57BL/6 mice, using the AE17 tumour model (Figures 4 B,E,H,I), which is relatively resistant to ICB and improved response. We also identified IFN β -positive cells in *Ifnb1^{tm1Lky}/J* mice which are on the C57BL/6 background.

2. Similarly, the authors used a narrow set of immunotherapies, and chose to treat all of their mice with anti-CTLA-4/anti-PD-L1, a combination that is not used clinically. In addition, CTLA-4 targeting therapies in mice appear to deplete Tregs, a function that does not clearly underly efficacy in humans. As with the comment above, this limitation just needs to be noted because it may limit the translatability of these findings.

Treatments blocking the PD1/PDL1 axis in combination with anti-CTLA4 have been tested and found to be a successful in multiple solid cancers and have been FDA approved in melanoma and non-small cell lung cancer (Chae YK, *J Immunother Cancer* 2018).

Indeed, anti-CTLA4 Clone 9H10 has been shown to deplete Tregs (Simpson *J Exp Med* 2013). However, our RNAseq data suggests Tregs are not depleted differentially between responders and non-responders. We added these graphs to figure S1D:

Flow cytometry of AE17 tumours 6 days after receiving the combination anti-CTLA4 and anti-PDL1 do not show a decrease in CD4+FoxP3+ cells:

Figure: Flow cytometry of tumours 6 days after ICB or PBS control, showing % FoxP3+ CD4+ T cell of CD45+ cells.

3. Overall, the data on RENCA cells are less convincing than with the other cell lines. This fact makes point #1 that much more important.

Although the kinetics between AB1 and Renca are indeed different, as discussed above, we validated our data experimentally in AB1, Renca and AE17, as well as using the human single cell data. In addition, as we discussed in the manuscript, independent groups in academia and industry have validated our findings in other tumour models and patient trials (e.g. Jerby-Arnon, *Cell* 2018; Lee, *JCI Insight* 2018; Ock *Nat Commun* 2017).

4. The human data are confusing and, at least for me, require more explanation. How precisely were the human “on/fast-off” signatures defined at a single time point. Were the authors looking at homologous genes that had been identified through the murine analysis, and presuming that they turned on a off, or did they use a pseudotime analysis? I am not sure that a pseudotime analysis could be considered valid here in any case, because the sample was taken prior to immunotherapy, and the authors demonstrated in mice that IFNbeta signaling should be on in tumors at the time of checkpoint blockade therapy, and that the process of turning off this pathway has to happen later in the treatment course to be beneficial. Simply indicating that IFN signaling is found in future responders to immunotherapy is not surprising, and is not specific enough to the authors mechanisms for them to consider this human sample validation.

Yes, we used the same fast-off IFN signature, and looked at homologous genes. The mapping between our fast-off gene set and homologous genes is provided in Data S5.

The human dataset was used as a validation dataset for the source of the IFN β on/off, demonstrating Ccr2⁺ monocytes are the predominant source of type I IFN activity. As we do not have access to longitudinal tumour samples from patients undergoing ICB with both responders and non-responders, simply because that’s not clinically and ethically feasible, we cannot map these dynamics in patients. This finding highlights monocytes as an important player in the ICB induced anti-tumour immune response, as has been reported in acute vs chronic viral infections before.

We also demonstrate that our fast-off signature has a higher normalised enrichment score when comparing CCR2⁺ myeloid cells between expanders and non-expanders and ranks statistically higher than the human dataset (ref. 25) gene set in pathway overdispersion analysis (Figure S10E, Data S7). We interpret these metrics to support the generalisability of our signature to human data.

5. Breast tumors also respond very poorly to immunotherapy in general, and so any findings in those tumors may be of limited value for understanding the immune response in tumors that respond more readily such as melanoma. I understand that the authors had these data available, but this limitation coupled with #4 should make them significantly temper their statements about validation in human samples, and I am not even convinced at this point that these data belong in this manuscript.

Clinical trials of anti PD-1 in PDL1+ TNBC have reported response rates of 18.5% – 23% (Nanda 2016 *J Clin Oncol*; Adams 2017 *J Clin Oncol*; Dirix 2017 *Breast Cancer Res Treat*). Atezolizumab (anti-PDL1) plus nab-paclitaxel chemotherapy was superior over chemotherapy alone for TNBC, with a response rate of 58.9%, including 10.7% complete responses, compared to 45.9% and 1.6%, respectively, in the chemotherapy only group (Schmid *NEJM* 2018). The breast cancer dataset from Bassez et al which we used are treatment naïve, early-stage cancer and therefore may indeed be relatively more responsive (9 out of 29 had T cell expansion), compared to those earlier monotherapy trials.

We tempered our statements about how the human data may support the IFN β dynamics, on page 9 and 11. The human data indicates a role for monocytic cells in T cell expansion after ICB.

I have a few small suggestions as well

6. Figure 3A really doesn't provide any information without having any of the genes labeled, which obviously won't be legible in a main figure. I suggest that this figure be moved to the supplement and that gene labels be added on a larger image.

Figure 3A gives an overview of the GENIE3 analysis, showing key regulators across all 144 samples in our time-course data. We use this analysis to survey regulator dynamics associated with the response to ICB. With this, we confirmed differential dynamic regulation between responders and non-responders to ICB. We therefore proceeded to focus on the top 10 known transcription factors (TF) that formed central hubs in the network. We agree, however, that additional information would be useful to our readers. In Data S4, we now provide the genes specific to each cluster and include gene set enrichment analysis for these gene sets.

7. Figure 3B and 3C should be enlarged – at the current size, I really cannot read what the authors are showing

The network figures in 3B and C show different activation of the separate modules (red is activated, blue is inhibited) in time. No other data can be extracted from this figure, but we feel there is value in displaying the dynamic expression profiles of the clusters, as they show the on/off signalling activity. We enlarged the figure.

8. On the survival curves, it would be nice to have the timing that treatments were given marked on the x-axis

We added those on figure 4. Dashes on x-axis represent day of dosing: black = poly(I:C) or rIFN, yellow = ICB, red = anti-IFN antibody.

9. Figure 4C and 4D may benefit from another method to validate the findings, as is the validity of the results are difficult to gauge – perhaps isolating the T cells or performing single cell sequencing at specific time points so that the authors are sure that the IFN response signal is indeed T cell derived

Please also see our reply to query 5 from reviewer 2. In figure 4C-D we used a gene signature from IFN β stimulated T cells (in comparison to other immune stimulants) which is the only comprehensive dataset of IFN β -stimulated cells currently available. This dataset is then applied to our bulk RNAseq data, and therefore is not T-cell specific as other immune cell types likely upregulate similar genes in response to IFN β . What this figure demonstrates is that the on/off signature is partially attributed to IFN β stimulation, opposed to IFN γ . We thank the reviewers for pointing out this was unclear and clarified it in the text on page 7 so the reader does not draw the conclusion it is restricted to T cells.

10. For figure 5I, it would be helpful to see the same plot for the YFP negative cells in order to assess the validity of the determined gates. A similar analysis should be coupled with the supplemental figure that shows the overall gating strategy.

We added this to the figure in the supplementary:

REVIEWERS' COMMENTS

Reviewer #1 (Remarks to the Author):

All my previous concerns were successfully addressed by the authors.

Reviewer #2 (Remarks to the Author):

I am quite satisfied that the authors have responded the concerns by performing large amount of additional in vitro and in vivo experiments. I have one additional suggestion :

The Fig S7 should be in the principal figure in my opinion. This experiment shows the clear-cut difference between chronic exposure and ON/fast-OFF exposure of IFN β : ON/fast-OFF exposure promotes the microenvironmental CD8+T cells into an activated state without further expression of exhaustion proteins (TIM3 or LAG3), and as one of the reviewers mentioned the problem of anti-CTLA4 which deplete Treg cells, in this experiment, they nicely showed chronic exposure and ON/fast-OFF exposure have similar effects on Treg, that I believe that, at least in their current models, Treg is not the main player. Whereas, FigS2 is not really necessary, because the analysis of checkpoint gene expression is based on bulk RNAseq, it is difficult to know the expression trends observed are due to which populations of cells in the tumor microenvironment. In addition, most of them have very similar trends in both responder and non-responder.

Reviewer #3 (Remarks to the Author):

The authors have submitted an improved manuscript where multiple experimental questions from this reviewer and the others were, in my opinion, adequately addressed. I do have several concerns about the model systems and human data that the authors use which impact the potential generalizability of the work.

Specific Comments

1. I did not previously fully appreciate the rationale for the models chosen. Nevertheless, the limited number of models, and the fact that they do not include the most commonly studied in immunotherapy, does reduce the potential generalizability of the findings. I can't see any reason why the tumor therapy experiments could not be repeated in more common tumor models (B16, CT26, MC38) to determine whether timed activation/inhibition of IFN β signaling is more generally important for optimal responses. Alternatively, the authors could acknowledge in their framing of the current manuscript that their findings may be limited to specific models.

2. While the authors are correct that the PD-1/PD-L1 axis has been targeted alongside CTLA-4 in patients and combination therapy has multiple approvals at this point, PD-L1 blockade is not approved in combination with CTLA-4 blockade. Although similar, these two strategies (anti-PD-L1/CTLA-4, anti-PD-1/CTLA-4) are not completely overlapping and this is a weakness in the presented studies, reducing the generalizability of the author's findings.

3. The fact that the authors do not observe a difference in Treg numbers by sequencing in responders versus non responders does not change the fact that 9H10 likely works through a different mechanism in mice than in humans (this has now been replicated by multiple groups in multiple models). The timing at which Tregs are assessed also matters for 9H10 and is model dependent, and the authors may have missed the window of depletion with their flow cytometry experiment, really the most accurate way to detect these cells. Regardless, the translatability of the findings remains unclear when the mechanism of action of one of the drugs studied is demonstrably different between the model system (mice) and humans. This isn't something that contradicts the findings of the study, but the authors should acknowledge these limitations more readily in the text. As it reads now, they draw conclusions that are more expansive about their findings than are warranted by their data.

4. Thank you for clarifying the purpose of the human data. I think that simply showing that the gene set is present in monocytes at a static point in time prior to immunotherapy and that this gene set correlates with T cell expansion in the tumor is interesting, but it is not sufficiently mechanistically tied to the author's hypothesis to really count as strong supportive data. As the analysis is currently worded, it could easily mislead a reader into thinking that the core mechanism the authors show in mice has been replicated in humans, which it really hasn't.

5. While difficult, obtaining longitudinal tumor samples is both clinically feasible and ethical if the patients are aware of the purpose of the research biopsies and consent to the process.

6. The trial results cited by the authors (Schmid NEJM 2018) was of marginal significance and authorization for the treatment was eventually withdrawn by the FDA after follow-up studies showed no overall survival benefit from immunotherapy. That said, the human data set that the authors used here is for neoadjuvant treatment, which has showed evidence of benefit, though with anti-PD-1 rather than anti-PD-L1 immunotherapy. Breast cancer has remained poorly immunogenic compared to multiple other tumor types, reducing the validity of assumptions of generalizability from mechanistic information from breast cancer responses. There is abundant evidence that tumor type matters in the mechanisms of immunotherapy response and resistance.

7. In the discussion, the authors discuss data that chronic IFN signaling is associated with resistance to immunotherapy, but do not discuss the data from patients that implicates mutations in IFN signaling in acquired resistance to immunotherapy, for example (PMID: 27433843, now replicated by this group and others).

REVIEWERS' COMMENTS

Reviewer #1 (Remarks to the Author):

All my previous concerns were successfully addressed by the authors.

Reviewer #2 (Remarks to the Author):

I am quite satisfied that the authors have responded the concerns by performing large amount of additional in vitro and in vivo experiments. I have one additional suggestion :

The Fig S7 should be in the principal figure in my opinion. This experiment shows the clear-cut difference between chronic exposure and ON/fast-OFF exposure of IFN β : ON/fast-OFF exposure promotes the microenvironmental CD8+T cells into an activated state without further expression of exhaustion proteins (TIM3 or LAG3), and as one of the reviewers mentioned the problem of anti-CTLA4 which deplete Treg cells, in this experiment, they nicely showed chronic exposure and ON/fast-OFF exposure have similar effects on Treg, that I believe that, at least in their current models, Treg is not the main player. Whereas, FigS2 is not really necessary, because the analysis of checkpoint gene expression is based on bulk RNAseq, it is difficult to know the expression trends observed are due to which populations of cells in the tumor microenvironment. In addition, most of them have very similar trends in both responder and non-responder.

We kindly thank the reviewer for this suggestion. We now added the Fig S7 data to the main Fig 4 and added a sentence underlining the point the reviewer makes on page 8: “[...], suggesting these are activated but not terminally exhausted T cells”. Although we fully agree with the reviewer that FACS data (in AE17) provide the most cellular granularity, we do feel that the data in Fig S2, showing differences in (particularly baseline) gene expression of several immune checkpoints (AB1/Renca), will be of relevance to the reader to assess the kinetics and the robustness of the findings across models.

Reviewer #3 (Remarks to the Author):

The authors have submitted an improved manuscript where multiple experimental questions from this reviewer and the others were, in my opinion, adequately addressed. I do have several concerns about the model systems and human data that the authors use which impact the potential generalizability of the work.

Specific Comments

1. I did not previously fully appreciate the rationale for the models chosen. Nevertheless, the limited number of models, and the fact that they do not include the most commonly studied in immunotherapy, does reduce the potential generalizability of the findings. I can't see any reason why the tumor therapy experiments could not be repeated in more common tumor models (B16, CT26, MC38) to determine whether timed activation/inhibition of IFN β signaling is more generally important for optimal responses. Alternatively, the authors could acknowledge in their framing of the current manuscript that their findings may be limited to specific models.

Thank you for the comment. To our knowledge, there is no data demonstrating that CT26, B16 or MC38 are most representative for human cancers, therefore providing more generalisable results. For example, the anti-tumour efficacy of CTLA4 blockade was identified in s.c. 51BLim10 colorectal cancer and SalN fibrosarcomas models (Leach, Science 1996); for anti-PD1 it was in s.c. P815 mastocytoma (Dong et al. Nat Med 2002); and for anti-PD-L1 in s.c. P815 mastocytoma and B16 melanoma (Iwai PNAS 2002). In later papers, those findings were expanded to other tumour models. We demonstrated a role for timed IFN β activation/inhibition in three independent mouse tumour models, across two strains (BALB/c and C57BL/6), including the widely used Renca model. We are looking forward to independent validation of the importance of the dynamics of IFN β signaling in other models and have added a sentence to the discussion on page 11: “We identified the requirement for on/off IFN β activity in the context of ICB in three mouse models, across two strains, but further future validation in other tumour models is warranted.”

2. While the authors are correct that the PD-1/PD-L1 axis has been targeted alongside CTLA-4 in patients and combination therapy has multiple approvals at this point, PD-L1 blockade is not approved in combination with CTLA-4 blockade. Although similar, these two strategies (anti-PD-L1/CTLA-4, anti-PD-1/CTLA-4) are not completely overlapping and this is a weakness in the presented studies, reducing the generalizability of the author's findings.

The data for combination PD-1 and CTLA-4 blockade is mature, given that this was the first combination checkpoint blockade therapy to be used successfully in clinical practice. The existence of multiple approvals for this combination is therefore unsurprising.

The reviewer correctly points out that PD-L1 blockade is currently not approved in combination with CTLA-4 blockade. Nevertheless, lack of approval does not imply lack of efficacy or lack of therapeutic equivalence to CTLA-4/PD-1 blockade, given that this combination is in the earlier stages of clinical validation. Of note, the combination of tremelimumab and durvalumab has been evaluated in multiple studies and shows clinical promise, and it is currently being considered by the FDA for hepatocellular carcinoma. Additionally, in cancers such as mesothelioma and hepatocellular carcinoma, both combination strategies show clinical efficacy, although no head-to-head comparisons of these two combinations has been performed.

Given the issues and uncertainties above, we feel that the question of generalisability can only be tackled after more clinical and translational data to becomes available to allow an objective comparison between the two treatment strategies. We eagerly await an opportunity to validate these results as the use of this CTLA-4/PD-L1 combination becomes more widespread in larger clinical studies.

3. The fact that the authors do not observe a difference in Treg numbers by sequencing in responders versus non responders does not change the fact that 9H10 likely works through a different mechanism in mice than in humans (this has now been replicated by multiple groups in multiple models). The timing at which Tregs are assessed also matters for 9H10 and is model dependent, and the authors may have missed the window of depletion with their flow cytometry experiment, really the most accurate way to detect these cells. Regardless, the translatability of the findings remains unclear when the mechanism of action of one of the drugs studied is demonstrably different between the model system (mice) and humans. This isn't something that contradicts the findings of the study, but the authors should acknowledge these limitations more readily in the text. As it reads now, they draw conclusions that are more expansive about their findings than are warranted by their data.

We measured intratumoural Tregs using both RNAseq (days 0-2-4-6 in AB1 and Renca) and flow cytometry (day 6 in AE17) and did not find any differences between responders and non-responders. Indeed, we cannot exclude a level of Treg depletion that went undetected by CIBERSORT and that rebounded to normal by day 6 as determined by flow cytometry, although we are not sure whether such a short and discrete depletion would be therapeutically relevant. We added a sentence to the text on page 8: "We did not observe any effect on Tregs but cannot fully exclude their involvement based on the literature regarding anti-CTLA4 in mice."

4. Thank you for clarifying the purpose of the human data. I think that simply showing that the gene set is present in monocytes at a static point in time prior to immunotherapy and that this gene set correlates with T cell expansion in the tumor is interesting, but it is not sufficiently mechanistically tied to the author's hypothesis to really count as strong supportive data. As the analysis is currently worded, it could easily mislead a reader into thinking that the core mechanism the authors show in mice has been replicated in humans, which it really hasn't.

Thank you for your comment. We rephrased the subheading as "On/fast-off IFN signature (instead of 'kinetics') in monocytic cells correlate with T-cell response in patients treated with ICB"

5. While difficult, obtaining longitudinal tumor samples is both clinically feasible and ethical if the patients are aware of the purpose of the research biopsies and consent to the process.

We do not dispute the feasibility and ethicality of longitudinal tumour biopsies. However, we would like to point out the clinical and logistical challenge of obtaining a validation dataset which would contain

tumour tissue sampled four times during immune checkpoint therapy, with 2-day intervals, prior to the initiation of the response, as we did in our animal cohorts. Longitudinal tumour samples are usually collected only at a pre-treatment and a post-treatment time point due to clinical constraints, thus providing only a fragmented description of dynamic tumour state. Additionally, the interval between pre-treatment and post-treatment samples varies between days to weeks, such that dynamic events lying outside this sampling interval will be missed.

6. The trial results cited by the authors (Schmid NEJM 2018) was of marginal significance and authorization for the treatment was eventually withdrawn by the FDA after follow-up studies showed no overall survival benefit from immunotherapy. That said, the human data set that the authors used here is for neoadjuvant treatment, which has showed evidence of benefit, though with anti-PD-1 rather than anti-PD-L1 immunotherapy. Breast cancer has remained poorly immunogenic compared to multiple other tumor types, reducing the validity of assumptions of generalizability from mechanistic information from breast cancer responses. There is abundant evidence that tumor type matters in the mechanisms of immunotherapy response and resistance.

Schmid et al. (2018) assessed chemotherapy in combination with immunotherapy for locally advanced and metastatic triple breast cancer. More recently, however, the FDA has approved immunotherapy in combination with chemotherapy for patients with metastatic triple negative breast cancer whose tumours expressed PD-L1 based on Keynote-355 data. In addition, Pembrolizumab has FDA approval both in the neo-adjuvant setting and for continuation in the adjuvant setting after surgery based on Keynote-522. Immunotherapy, therefore, has a role in treatment of breast cancer.

We emphasize that our analysis does not attempt to generalise from murine data to the mechanistic elements of the immunotherapy response specific to breast cancer. Our analysis focuses specifically on whether there is a discernible interferon signature analogous to the dynamic signature found in our murine data in immunotherapy treated patient tumours who display a beneficial response to therapy (in this case T cell expansion). We have amended the text in the manuscript to better reflect this, on page 12: "Although these patient samples do not provide the temporal granularity of four time points or definitive clinical response data, like our murine study does, which limits the ability to identify kinetic mechanisms underlying ICB efficacy, they do highlight the contribution of monocytes in the response to ICB."

7. In the discussion, the authors discuss data that chronic IFN signaling is associated with resistance to immunotherapy, but do not discuss the data from patients that implicates mutations in IFN signaling in acquired resistance to immunotherapy, for example (PMID: 27433843, now replicated by this group and others).

The mentioned paper identified mutations that led to inhibited IFN-gamma signalling in 4 melanoma patients with secondary resistance to ICB. This is in line with our findings showing that blocking IFN-gamma reduced the therapeutic effect of ICB. We added a sentence and the reference to the text in the relevant section on page 7: "In both models, the beneficial effect of blocking type I IFN after administration was negated by blocking type II IFN simultaneously, which is in line with reports in patients with tumour defects in IFN γ signalling associated with acquired resistance to ICB."